# Local Manifold Approximation and Projection
# for Manifold-Aware Diffusion Planning

**Kyowoon Lee** [1]  **Jaesik Choi** [1 2]

## Abstract

Recent advances in diffusion-based generative modeling have demonstrated significant promise in tackling long-horizon, sparse-reward tasks by leveraging offline datasets. While these approaches have achieved promising results, their reliability remains inconsistent due to the inherent stochastic risk of producing infeasible trajectories, limiting their applicability in safety-critical applications. We identify that the primary cause of these failures is inaccurate guidance during the sampling procedure, and demonstrate the existence of manifold deviation by deriving a lower bound on the guidance gap. To address this challenge, we propose *Local Manifold Approximation and Projection* (LoMAP), a *training-free* method that projects the guided sample onto a low-rank subspace approximated from offline datasets, preventing infeasible trajectory generation. We validate our approach on standard offline reinforcement learning benchmarks that involve challenging long-horizon planning. Furthermore, we show that, as a standalone module, LoMAP can be incorporated into the hierarchical diffusion planner, providing further performance enhancements.

## 1. Introduction

Planning over long horizons is crucial for autonomous systems operating in real-world settings, where rewards are sparse and actions often entail delayed consequences (Lee et al., 2023a). When environment dynamics are fully known, methods such as Model Predictive Control (Tassa et al., 2012) and Monte Carlo Tree Search (Silver et al., 2016; 2017; Lee et al., 2018) have achieved remarkable success. In most practical applications, however, the dynamics are not readily available and must be learned from data. Model-based reinforcement learning (MBRL) (Sutton, 2018) addresses this by coupling learned dynamics models with planners, offering increased data-efficiency and the flexibility to adapt across tasks. While MBRL offers a promising framework, it is vulnerable to adversarial plan exploitation when the learned model is imperfect (Talvitie, 2014; Asadi et al., 2018; Luo et al., 2019; Janner et al., 2019; Voelcker et al., 2022).

Recent progress in diffusion models provides an appealing alternative for long-horizon planning. Originally introduced as powerful generative models that iteratively reverse a multistep noising process (Ho et al., 2020; Song et al., 2021), diffusion models have demonstrated state-of-the-art sample quality across various domains (Nichol et al., 2022; Luo & Hu, 2021; Li et al., 2022). Building upon these successes, several works (Janner et al., 2022; Ajay et al., 2023; Liang et al., 2023) have leveraged diffusion to model entire trajectories in sequential decision-making tasks. By avoiding step-by-step autoregressive prediction, these approaches reduce error accumulation and naturally capture long-range dependencies. Moreover, by leveraging guided sampling (Dhariwal & Nichol, 2021), diffusion planners can sample trajectories biased toward high-return behaviors, achieving notable performance on standard offline reinforcement learning (RL) benchmarks (Fu et al., 2020).

Despite these advantages, diffusion planners struggle to guarantee reliable and feasible plans due to their inherent *stochasticity*. Unlike deterministic models that produce consistent outputs for a given input, diffusion models generate probabilistic trajectories. While this enables diverse sampling, it introduces the risk of generating physically implausible trajectories, which are referred to as *artifacts* in vision domains (Bau et al., 2019; Shen et al., 2020). Furthermore, the reward-guided sampling procedure used by unconditional diffusion planners, which jointly defines the sampling distribution and the energy function, can be hard to estimate accurately (Lu et al., 2023b). This inaccuracy may cause intermediate trajectory samples far away from the underlying data manifold during denoising, ultimately producing infeasible or low-quality trajectories and preclud-

[1]Korea Advanced Institute of Science and Technology (KAIST) [2]INEEJI. Correspondence to: Jaesik Choi <jaesik.choi@kaist.ac.kr>.

*Proceedings of the 42$^{nd}$ International Conference on Machine Learning*, Vancouver, Canada. PMLR 267, 2025. Copyright 2025 by the author(s).

Codes are available at github.com/leekwoon/lomap.

ing planners from being useful in safetycritical applications. To address these limitations, recent works (Lee et al., 2023b; Feng et al., 2024) introduced trajectory-refinement strategies to enhance sample quality, though the challenge of ensuring fully reliable plans remains an open problem.

In this paper, we demonstrate that manifold deviation occurs during diffusion sampling by establishing a lower bound on the guidance estimation error. To address this challenge, we introduce **Lo**cal **M**anifold **A**pproximation and **P**rojection (**LoMAP**), a *training-free* method that projects guided samples back onto a low-rank subspace approximated from offline datasets. LoMAP operates entirely at test time, requiring no additional training. At each reverse-diffusion step, it retrieves a few offline trajectories closest to the *denoised* version of the current sample, forward-diffuses these neighbors, and applies principal component analysis (PCA) to span a local low-dimensional subspace. The current sample is then *projected* onto this subspace, thereby significantly reducing off-manifold deviations and improving the likelihood of generating valid behaviors. Because LoMAP only adds a simple projection step after each reward-guided update, it is easily integrated into existing diffusion planners and effectively prevents manifold deviation.

Our main contributions are as follows: **(1)** We illustrate the manifold deviation issue in diffusion planners by deriving a theoretical lower bound on the guidance estimation error. **(2)** We propose *Local Manifold Approximation and Projection* (LoMAP), a training-free, plug-and-play module for diffusion planners that mitigates manifold deviation through local low-rank projections. **(3)** We demonstrate the effectiveness of LoMAP on standard offline RL benchmarks, particularly in challenging AntMaze task.

## 2. Background

### 2.1. Problem Setting

We consider a Markov decision process (MDP) described by the tuple $\langle \mathcal{S}, \mathcal{A}, P, r, \gamma \rangle$. Here, $\mathcal{S}$ is the state space, $\mathcal{A}$ is the action space, $P : \mathcal{S} \times \mathcal{A} \times \mathcal{S} \to [0, +\infty)$ is the transition model, $r : \mathcal{S} \times \mathcal{A} \to \mathbb{R}$ is a reward function, and $\gamma \in [0, 1]$ is the discount factor. Given a planning horizon $T$, the objective of trajectory optimization is to find the action sequence $\boldsymbol{a}_{0:T}^*$ that maximizes the expected return:

$$\boldsymbol{a}_{0:T}^* = \arg\max_{\boldsymbol{a}_{0:T}} \mathcal{J}(\boldsymbol{\tau}) = \arg\max_{\boldsymbol{a}_{0:T}} \sum_{t=0}^{T} \gamma^t r(\boldsymbol{s}_t, \boldsymbol{a}_t),$$

where $\boldsymbol{\tau} = (\boldsymbol{s}_0, \boldsymbol{a}_0, \boldsymbol{s}_1, \boldsymbol{a}_1, \ldots, \boldsymbol{s}_T, \boldsymbol{a}_T)$ is a trajectory, and $\mathcal{J}(\boldsymbol{\tau})$ represents the expected return of the trajectory.

### 2.2. Planning with Diffusion Models

Diffusion planners (Janner et al., 2022) utilize diffusion probabilistic models (Sohl-Dickstein et al., 2015; Ho et al., 2020) to model a trajectory distribution as a Markov chain with Gaussian transitions:

$$p_\theta(\boldsymbol{\tau}^0) = \int p(\boldsymbol{\tau}^M) \prod_{i=1}^{M} p_\theta(\boldsymbol{\tau}^{i-1}|\boldsymbol{\tau}^i) \, \mathrm{d}\boldsymbol{\tau}^{1:M} \quad (1)$$

where $p(\boldsymbol{\tau}^M)$ is a standard Gaussian prior, $\boldsymbol{\tau}^0$ is a noise-free trajectory, and $p_\theta(\boldsymbol{\tau}^{i-1}|\boldsymbol{\tau}^i)$ is a denoising process which is a learnable Gaussian transition:

$$p_\theta(\boldsymbol{\tau}^{i-1}|\boldsymbol{\tau}^i) = \mathcal{N}(\boldsymbol{\tau}^{i-1}|\boldsymbol{\mu}_\theta(\boldsymbol{\tau}^i), \boldsymbol{\Sigma}^i). \quad (2)$$

This reverse process inverts a forward process that incrementally corrupts the data with Gaussian noise according to a variance schedule $\{\beta_i\}_{i=1}^M$:

$$q(\boldsymbol{\tau}^i|\boldsymbol{\tau}^{i-1}) := \mathcal{N}(\boldsymbol{\tau}^i; \sqrt{1 - \beta_i}\boldsymbol{\tau}^{i-1}, \beta_i \boldsymbol{I}). \quad (3)$$

One useful property is the ability to directly sample $\boldsymbol{\tau}^i$ from $\boldsymbol{\tau}^0$ at any diffusion timestep $i$:

$$q(\boldsymbol{\tau}^i \mid \boldsymbol{\tau}^0) = \mathcal{N}(\boldsymbol{\tau}^i; \sqrt{\alpha_i}\boldsymbol{\tau}^0, (1 - \alpha_i)\boldsymbol{I}), \quad (4)$$

where $\alpha_i := \prod_{s=1}^{i}(1 - \beta_s)$. The variance schedule is designed to respect $\alpha_M \approx 0$ so that $\boldsymbol{\tau}^M$ becomes close to $\mathcal{N}(\boldsymbol{0}, \boldsymbol{I})$.

During the training procedure, rather than directly parameterizing $\boldsymbol{\mu}_\theta$, Diffuser trains noise-predictor model $\boldsymbol{\epsilon}_\theta$ to predict the noise $\boldsymbol{\epsilon}$ that was added to corrupt $\boldsymbol{\tau}^0$ into $\boldsymbol{\tau}^i$ (Ho et al., 2020):

$$\mathcal{L}(\theta) := \mathbb{E}_{i,\boldsymbol{\epsilon},\boldsymbol{\tau}^0}[\|\boldsymbol{\epsilon} - \boldsymbol{\epsilon}_\theta(\boldsymbol{\tau}^i)\|^2], \quad (5)$$

where $\boldsymbol{\tau}^i = \sqrt{\alpha_i}\boldsymbol{\tau}^0 + \sqrt{1 - \alpha_i}\boldsymbol{\epsilon}$ and $\boldsymbol{\epsilon} \sim \mathcal{N}(\boldsymbol{0}, \boldsymbol{I})$.

**Trajectory optimization as guided sampling.** To generate trajectories that have high returns, the energy-guided sampling can be considered:

$$\tilde{p}_\theta(\boldsymbol{\tau}^0) \propto p_\theta(\boldsymbol{\tau}^0) \exp(\mathcal{J}(\boldsymbol{\tau}^0)), \quad (6)$$

Therefore, Diffuser trains a separate regression network $\mathcal{J}_\phi$ that predicts the return $\mathcal{J}(\boldsymbol{\tau}^0)$ based on a noisily perturbed version $\boldsymbol{\tau}^i$. This is accomplished through the mean-square-error (MSE) objective:

$$\min_\phi \mathbb{E}_{i,\boldsymbol{\epsilon},\boldsymbol{\tau}^0} \left[ \|\mathcal{J}_\phi(\boldsymbol{\tau}^i) - \mathcal{J}(\boldsymbol{\tau}^0)\|_2^2 \right]. \quad (7)$$

During the sampling stage, a classifier guidance (Dhariwal & Nichol, 2021) is employed, incorporating gradients of $\mathcal{J}_\phi$ into the reverse diffusion process in Eq. (2). Specifically, the mean is updated as follows:

$$\tilde{p}_\theta(\boldsymbol{\tau}^{i-1}|\boldsymbol{\tau}^i) = \mathcal{N}(\boldsymbol{\tau}^{i-1}|\boldsymbol{\mu}_\theta(\boldsymbol{\tau}^i) + \omega\boldsymbol{\Sigma}^i g, \boldsymbol{\Sigma}^i), \quad (8)$$

where $g = \nabla_{\boldsymbol{\tau}} \mathcal{J}_\phi(\boldsymbol{\tau})|_{\boldsymbol{\tau}=\boldsymbol{\mu}_\theta(\boldsymbol{\tau}^i)}$ and $\omega$ is the guidance scale that controls the strength of the guidance.

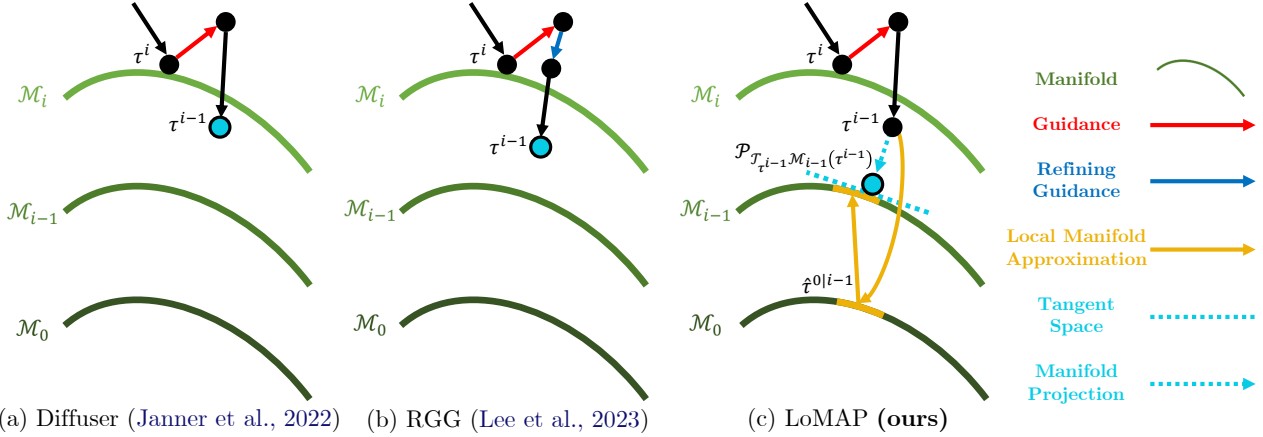

(a) Diffuser (Janner et al., 2022)    (b) RGG (Lee et al., 2023)    (c) LoMAP **(ours)**

Figure 1: A schematic overview of our approach, contrasted with Diffuser (Janner et al., 2022) and RGG (Lee et al., 2023b). As described in Section 3.1, *inexact* guidance arises in high-dimensional settings, causing deviations from the data manifold. RGG addresses this by refining samples via an OOD detection metric but relies on finely tuned guidance steps. In contrast, our LoMAP framework projects guided samples back onto a local low-rank subspace (Section 3.2), ensuring that sampling remains closer to the data manifold at each diffusion step.

## 2.3. Tweedie's Formula for Denoising

When samples are perturbed by Gaussian noise $\tilde{\tau} \sim \mathcal{N}(\tau, \sigma^2 \mathbf{I})$, Tweedie's formula (Robbins, 1992) provides a Bayes-optimal denoised estimate for observations:

$$\mathbb{E}[\tau|\tilde{\tau}] = \tilde{\tau} + \sigma^2 \nabla_{\tilde{\tau}} \log p(\tilde{\tau}). \tag{9}$$

where $p(\tilde{\tau}) \coloneqq \int p(\tilde{\tau}|\tau)p(\tau)\mathrm{d}\tau$. If we consider a discrete-time diffusion model with perturbation in Eq. (4), we can get the posterior mean by rewriting Tweedie's fomula (Chung et al., 2023; 2022):

$$
\begin{aligned}
\mathbb{E}[\tau^0|\tau^i] &= \frac{1}{\sqrt{\alpha_i}}\big(\tau^i + (1 - \alpha_i) \nabla_{\tau^i} \log p(\tau^i)\big) \\
&\approx \frac{1}{\sqrt{\alpha_i}}\big(\tau^i - \sqrt{1 - \alpha_i}\,\epsilon_\theta(\tau^i)\big),
\end{aligned}
\tag{10}
$$

where the scaled score function is estimated by $\epsilon_\theta$: $\nabla_{\tau^i} \log p(\tau^i) \approx -\epsilon_\theta(\tau^i)/\sqrt{1 - \alpha_i}$.

## 2.4. Low-dimensional Manifold Assumption

High-dimensional trajectory often exhibits intrinsic low-dimensional structure. We formalize this through the following assumption:

**Assumption 2.1.** (Low-dimensional Manifold Assumption). The set of clean data $\mathcal{M}_0$ lies on a $k$-dimensional subspace $\mathbb{R}^k$ with $k \ll d$.

Under this assumption, recent study (Chung et al., 2022) has shown that the set of noisy data $\tau^i$ is inherently concentrated around a $(d - k)$ dimensional manifold $\mathcal{M}_i$.

## 3. Manifold-Aware Diffusion Planning

In this section, we formalize the phenomenon of *manifold deviation*, a critical limitation of diffusion planners caused by inexact guidance (Section 3.1), and propose *Local Manifold Approximation and Projection* (LoMAP), a training-free method to preserve trajectory feasibility (Section 3.2).

### 3.1. Manifold Deviation by Inexact Guidance

Recall from Equation (6) that diffusion planners aim to sample trajectories biased toward those that have high returns by sampling from the following *energy-guided* distribution:

$$\tilde{p}_\theta(\tau^0) \;\propto\; p_\theta(\tau^0) \exp[\mathcal{J}(\tau^0)],$$

where $p_\theta(\tau^0)$ is the learned trajectory distribution from the diffusion model, and $\mathcal{J}(\tau^0)$ is the (negative) energy or return function to be maximized.

Following Theorem 3.1 in (Lu et al., 2023b), consider the forward process $q(\tau^i \mid \tau^0)$. Then the marginal distribution at diffusion timestep $i$ is:

$$
\begin{aligned}
\tilde{p}_\theta(\tau^i) &= \int q(\tau^i|\tau^0)\tilde{p}_\theta(\tau^0)\,\mathrm{d}\tau^0 \\
&= \int q(\tau^i|\tau^0)p_\theta(\tau^0)\frac{e^{\mathcal{J}(\tau^0)}}{Z}\,\mathrm{d}\tau^0 \\
&= p_\theta(\tau^i)\int q(\tau^0|\tau^i)\frac{e^{\mathcal{J}(\tau^0)}}{Z}\,\mathrm{d}\tau^0 \\
&= \frac{p_\theta(\tau^i)\mathbb{E}_{q(\tau^0|\tau^i)}\left[e^{\mathcal{J}(\tau^0)}\right]}{Z} \\
&\propto p_\theta(\tau^i) \exp\big[\underbrace{\log \mathbb{E}_{q(\tau^0|\tau^i)}\left[e^{\mathcal{J}(\tau^0)}\right]}_{\mathcal{J}_t(\tau^i)}\big].
\end{aligned}
$$

**Algorithm 1 Manifold-Aware Diffusion Planning**

1: **Require** Diffuser $\mu_\theta$, guide $\mathcal{J}_\phi$, covariances $\Sigma^i$, scale $\omega$, offline dataset $\{\tau_n^0\}_{n=1}^N$, number of neighbors $k$
2: **while** not done **do**
3:     Observe state $s$; initialize plan $\tau^N \sim \mathcal{N}(\mathbf{0}, \mathbf{I})$
4:     **for** $i = N, \ldots, 1$ **do**
5:         // parameters of reverse transition
6:         $\mu \leftarrow \mu_\theta(\tau^i)$
7:         // guide using gradients of return
8:         $\tau^{i-1} \sim \mathcal{N}\big(\mu + \omega\,\Sigma^i\,\nabla_\mu \mathcal{J}_\phi^{\text{MSE}}(\mu), \Sigma^i\big)$
9:         // manifold projection (LoMAP)
10:       $\tau^{i-1} \leftarrow \text{LoMAP}\big(\tau^{i-1}; \{\tau_n^0\}, k\big)$
11:       // constrain first state of plan
12:       $\tau_{s_0}^{i-1} \leftarrow s$
13:     **end for**
14:     Execute first action of plan $\tau_{a_0}^0$
15: **end while**

**Algorithm 2 LoMAP$(\tau^{i-1}, \{\tau_n^0\}, k)$**

1: **Input:** Noisy trajectory sample $\tau^{i-1}$, offline dataset $\{\tau_n^0\}_{n=1}^N$, number of neighbors $k$
2: **Output:** Projected sample $\tau^{i-1}$
3: // denoise the current sample
4: $\hat{\tau}^{\,0|(i-1)} \leftarrow \frac{1}{\sqrt{\alpha_{i-1}}}\Big(\tau^{i-1} - \sqrt{1 - \alpha_{i-1}}\,\epsilon_\theta(\tau^{i-1})\Big)$
5: // $k$ nearest neighbors
6: $\mathcal{N} \leftarrow \text{TopKNeighbors}\big(\hat{\tau}^{\,0|(i-1)}; \{\tau_n^0\}, k\big)$
7: // forward-diffuse neighbors
8: **for** $\tau_{(n_j)}^0 \in \mathcal{N}$ **do**
9:     $\tau_{(n_j)}^{i-1} \leftarrow \sqrt{\alpha_{i-1}}\,\tau_{(n_j)}^0 + \sqrt{1 - \alpha_{i-1}}\,\epsilon_{(n_j)}$
10: **end for**
11: // PCA on the local neighborhood
12: $\mathbf{U} \leftarrow \text{PCA}\big(\{\tau_{(n_j)}^{i-1}\}\big)$
13: // project onto local subspace
14: $\tau^{i-1} \leftarrow \mathbf{U}\,\mathbf{U}^\top\,\tau^{i-1}$
15: **return** $\tau^{i-1}$

---

Hence, the *exact* intermediate guidance at diffusion timestep $i$ is given by the gradient of

$$\mathcal{J}_t(\tau^i) = \log \mathbb{E}_{q(\tau^0|\tau^i)}\big[\exp(\mathcal{J}(\tau^0))\big].$$

By injecting this exact guidance into each reverse diffusion step, the sampled $\tau^0$ exactly follows the desired distribution Eq. (6) by rewriting the score of

$$\nabla_{\tau^i} \log \tilde{p}_\theta(\tau^i) = \underbrace{\nabla_{\tau^i} \log p_\theta(\tau^i)}_{\approx -\epsilon_\theta(\tau^i)/\sqrt{1-\alpha_i}} - \underbrace{\nabla_{\tau^i} \mathcal{J}_t(\tau^i)}_{\text{guidance}} \quad (11)$$

where the first term is approximated by the learned noise-predictor model $\epsilon_\theta$.

**Guidance Gap.** In practice, however, many existing diffusion planners (Janner et al., 2022; Liang et al., 2023; Chen et al., 2024) learn an approximate guidance function $\mathcal{J}_\phi^{\text{MSE}}(\tau^i)$ via mean-square-error (MSE) objective:

$$\min_\phi \mathbb{E}_{i,\epsilon,\tau^0}\left[\|\mathcal{J}_\phi(\tau^i) - \mathcal{J}(\tau^0)\|_2^2\right], \quad (12)$$

where $\tau^i = \sqrt{\alpha_i}\tau^0 + \sqrt{1-\alpha_i}\epsilon$ and $\epsilon \sim \mathcal{N}(\mathbf{0}, \mathbf{I})$.

However, with sufficient model capacity, the optimal $\mathcal{J}_\phi$ under the MSE objective satisfies:

$$\mathcal{J}_\phi^{\text{MSE}}(\tau^i) = \mathbb{E}_{q(\tau^0|\tau^i)}\big[\mathcal{J}(\tau^0)\big]$$
$$\leq \log \mathbb{E}_{q(\tau^0|\tau^i)}\big[e^{\mathcal{J}(\tau^0)}\big] = \mathcal{J}_t(\tau^i),$$

where the inequality follows from Jensen's inequality, implying that $\mathcal{J}_\phi^{\text{MSE}}$ *underestimates* the desired quantity.

**Definition 3.1.** (Guidance gap). Let $\nabla_{\tau^i} \mathcal{J}_t(\tau^i)$ denote the *true* intermediate guidance at diffusion step $i$, and let

$\nabla_{\tau^i} \mathcal{J}_\phi^{\text{MSE}}(\tau^i)$ be the *estimated guidance* via an MSE-based objective. We define the *guidance gap* at $\tau^i$ by

$$\Delta_{\text{guidance}}(\tau^i) = \big\|\nabla_{\tau^i} \mathcal{J}_t(\tau^i) - \nabla_{\tau^i} \mathcal{J}_\phi^{\text{MSE}}(\tau^i)\big\|_2. \quad (13)$$

To study how inaccuracies in energy guidance grow with dimensionality, we introduce the *guidance gap* in Eq. (13). Proposition 3.2 shows that this gap has a lower bound on the order of $\sqrt{d}$ in high-dimensional regimes.

**Proposition 3.2.** *(Dimensional scaling of guidance gap.) Suppose $\mathcal{J}(\tau^0)$ is not constant. Given the* true *guidance*

$$\nabla_{\tau^i} \mathcal{J}_t(\tau^i) = \frac{\mathbb{E}_{q(\tau^0|\tau^i)}\big[e^{\mathcal{J}(\tau^0)}\nabla_{\tau^i}\log q(\tau^0|\tau^i)\big]}{\mathbb{E}_{q(\tau^0|\tau^i)}\big[e^{\mathcal{J}(\tau^0)}\big]},$$

*and the* MSE-based *guidance*

$$\nabla_{\tau^i} \mathcal{J}_\phi^{\text{MSE}}(\tau^i) = \mathbb{E}_{q(\tau^0|\tau^i)}\big[\mathcal{J}(\tau^0)\nabla_{\tau^i}\log q(\tau^0|\tau^i)\big],$$

*there exists a choice of $\tau^i$ such that*

$$\left\|\nabla_{\tau^i} \mathcal{J}_t(\tau^i) - \nabla_{\tau^i} \mathcal{J}_\phi^{\text{MSE}}(\tau^i)\right\|_2 \geq \frac{c}{\sqrt{1-\alpha_i}}\sqrt{d},$$

*for some constant $c > 0$ that does not depend on $d$.*

*Proof Sketch.* The difference between the true guidance and the MSE-based guidance can be expressed as an expectation involving $\delta(\tau^0) := \left(\frac{e^{\mathcal{J}(\tau^0)}}{\mathbb{E}[e^{\mathcal{J}(\tau^0)}]} - \mathcal{J}(\tau^0)\right)$ times the forward-process noise $\epsilon$. By Jensen's inequality, $\delta(\tau^0)$ has positive mean, indicating that the MSE-based guidance underestimates the exponential-weighted return. Exploiting

the typical behavior $\|\epsilon\|_2 \approx \sqrt{d}$ in high dimensions and choosing $\tau^i$ so that $\delta$ aligns well with $\epsilon$, we derive the guidance gap scaling on the order of $\sqrt{d}$. For the complete proof, see Appendix A. □

Consequently, as indicated by Proposition 3.2, this issue becomes more severe in scenarios involving long planning horizons and high-dimensional state and action spaces. The substantial guidance gap forces $\tau^{i-1}$ to drift from the intermediate data manifold $\mathcal{M}_{i-1}$, leading sampled trajectories away from the feasible manifold a problem we refer to as *manifold deviation*. Figure 4 provides empirical evidence of this issue.

### 3.2. Local Manifold Approximation and Projection

As explained in Section 3.1, inaccuracies in the energy-guided update can cause the sample $\tau^i$ to deviate from the underlying data manifold. To mitigate manifold deviation caused by inexact guidance, we propose **LoMAP** - a training-free method that projects guided samples back to the data manifold through local low-rank approximations. The key insight is that while intermediate diffusion samples $\tau^i$ may deviate from the manifold, their *denoised estimates* can guide local manifold approximation using the offline dataset.

**Manifold-aware guidance.** Given a trajectory sample $\tau^i$, we sample $\tau^{i-1}$ with *manifold-aware guidance* in two steps:

$$\tau^{i-1} \sim \mathcal{N}\Big( \mu_\theta(\tau^i) + \omega\, \Sigma^i\, g,\ \Sigma^i \Big), \tag{14}$$

$$\tau^{i-1} \leftarrow \mathcal{P}_{\mathcal{T}_{\tau^{i-1}} \mathcal{M}_{i-1}}\big(\tau^{i-1}\big), \tag{15}$$

where $g = \nabla_{\tau^{i-1}} \mathcal{J}_\phi^{\mathrm{MSE}}(\tau^{i-1})$ is the gradient-based guidance term, $\omega$ is the guidance scale, and $\mathcal{P}_{\mathcal{T}_{\tau^{i-1}} \mathcal{M}_{i-1}}$ denotes projection onto the local manifold. Eq. (14) applies a reward-guided shift to sample $\tau^{i-1}$, while Eq. (15) *projects* $\tau^{i-1}$ onto the low-dimensional subspace derived from the offline dataset, mitigating drift away from feasible trajectories.

**Approximating the local manifold.** We estimate $\mathcal{T}_{\tau^{i-1}} \mathcal{M}_{i-1}$ using a *local* low-rank approximation from the offline dataset of feasible trajectories. To mitigate noise, we first form a denoised surrogate

$$\hat{\tau}^{0|i-1} = \frac{1}{\sqrt{\alpha_{i-1}}}\big(\tau^{i-1} - \sqrt{1-\alpha_{i-1}}\,\epsilon_\theta(\tau^{i-1})\big),$$

using Tweedie's formula (Eq. 9), where $\epsilon_\theta$ is the trained noise-prediction network. We then retrieve $k$ nearest neighbors of $\hat{\tau}^{0|i-1}$ from the *clean* offline trajectories, $\{\tau_{(n_j)}^0\}_{j=1}^k$, using cosine similarity in trajectory space following (Feng et al., 2024). Next, we *forward diffuse* these clean neighbors to timestep $i-1$:

$$\tau_{(n_j)}^{i-1} = \sqrt{\alpha_{i-1}}\,\tau_{(n_j)}^0 + \sqrt{1-\alpha_{i-1}}\,\epsilon_{(n_j)}, \quad \epsilon_{(n_j)} \sim \mathcal{N}(\mathbf{0}, \mathbf{I}).$$

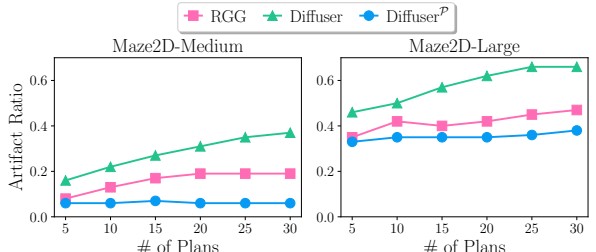

Figure 2: Artifact ratios on Maze2D-Medium (left) and Maze2D-Large (right) as the number of sampled plans increases. The y-axis denotes the fraction of trajectories that pass through walls, making them infeasible. Across both tasks, our LoMAP augmented Diffuser$^\mathcal{P}$ consistently produces fewer artifact plans compared to Diffuser and RGG.

Because each $\tau_{(n_j)}^{i-1}$ remains close to the manifold at timestep $i-1$, these $k$ samples approximate the local neighborhood $\mathcal{M}_{i-1}$. We then perform a rank-$r$ PCA on $\{\tau_{(n_j)}^{i-1}\}_{j=1}^k$ to obtain an orthonormal basis $U \in \mathbb{R}^{d \times r}$. The matrix $U$ spans an $r$-dimensional subspace that approximates $\mathcal{T}_{\tau^{i-1}}\mathcal{M}_{i-1}$. Thus,

$$\mathcal{P}_{\mathcal{T}_{\tau^{i-1}} \mathcal{M}_{i-1}}(\mathbf{z}) = U\,U^\top\,\mathbf{z},$$

which retains only the principal directions of variation supported by the offline data. In practice, $r \ll d$, and we choose $r$ by retaining the principal components that explain at least a fraction $\lambda$ of the total variance. In practice, we find that setting $\lambda = 0.99$ works well. Pseudocode for the manifold-aware planning method is provided in Algorithm 1. Notably, our LoMAP module is entirely training-free and can be readily integrated into existing diffusion planners by simply adding a manifold-projection step after each reward-guided update. For implementation details, including efficient manifold approximation and projection, see Appendix F.

## 4. Experiments

In this section, we present experimental results showing that augmenting prior diffusion planners with LoMAP improves planning performance across a variety of offline control tasks. Specifically, we demonstrate **(1)** that LoMAP effectively mitigates manifold deviation and filters out artifact trajectories, **(2)** that it further enhances planning performance when integrated into diffusion planner, and **(3)** that LoMAP, as a plug-and-play module, can be seamlessly incorporated into hierarchical diffusion planners, enabling successful planning in the challenging AntMaze domain. Additional details regarding our experimental setup and implementation are provided in Appendix E.

### 4.1. Mitigating Manifold Deviation

To investigate whether LoMAP effectively mitigates manifold deviation in diffusion-based planners, we apply it to Dif-

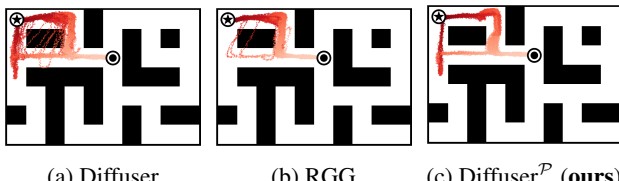

| (a) Diffuser | (b) RGG | (c) Diffuser$^{\mathcal{P}}$ (**ours**) |

Figure 3: Visualization of 100 sampled trajectories from Diffuser, RGG, and Diffuser$^{\mathcal{P}}$ in Maze2D, under a specified start ◉ and goal ✪ condition.

fuser (Janner et al., 2022) and refer to the resulting method as Diffuser$^{\mathcal{P}}$. We conduct a quantitative evaluation of manifold deviation in Maze2D tasks by leveraging an oracle that identifies *artifact plans*, defined as trajectories that pass through walls and are thus physically infeasible. We compare Diffuser$^{\mathcal{P}}$ against Diffuser and a baseline variant, Restoration Gap Guidance (RGG) (Lee et al., 2023b), in tasks where the planner is given only a start and goal location. Specifically, we randomly select start and goal states and generate trajectories under increasing sample sizes. For each start-goal pair, if at least one sampled trajectory contains invalid transitions through walls, we mark that pair as exhibiting manifold deviation. As shown in Figure 2, Diffuser produces some infeasible transitions, especially in the more complex Maze2D-Large environment. Although RGG partially alleviates this issue, Diffuser$^{\mathcal{P}}$ demonstrates the highest reliability, consistently generating valid trajectories even when a larger number of plans are drawn.

This phenomenon is further illustrated in Figure 3. Although RGG removes many artifact plans, it also reduces the diversity of solutions, clustering trajectories near a narrower set of paths. By contrast, Diffuser$^{\mathcal{P}}$ maintains high reliability and diversity, producing physically feasible trajectories without sacrificing coverage of the solution space.

### 4.2. Enhancing Planning Performance

**Maze2D.** To further demonstrate how LoMAP improves planning performance, we evaluate it on Maze2D environments (Fu et al., 2020), which involve navigating an agent to a target goal location through complex mazes requiring long-horizon planning. Maze2D features two distinct tasks: a single-task setup where the goal location is fixed, and a multi-task variant (Multi2D) in which the goal is randomized at the start of each episode. We compare our methods against the model-free offline RL algorithm IQL (Kostrikov et al., 2022) and two trajectory-refinement approaches for diffusion planners, RGG (Lee et al., 2023b) and TAT (Feng et al., 2024).

As shown in Table 1, the model-free IQL suffers a notable performance drop under multi-task conditions, likely due to the challenges of credit assignment. By contrast, diffusion-based planners perform well in both single-task and multi-

Table 1: Comparison on Maze2D for Diffuser$^{\mathcal{P}}$, Diffuser, and prior methods. Diffuser$^{\mathcal{P}}$ denotes Diffuser augmented with LoMAP. We report the mean and the standard error over 1000 planning seeds.

| Environment | | IQL | RGG | TAT | Diffuser | Diffuser$^{\mathcal{P}}$ |
|---|---|---|---|---|---|---|
| Maze2d | U-Maze | 47.4 | 108.8 | 114.5 | 113.9 | **126.0**$_{\pm 0.26}$ |
| | Medium | 34.9 | **131.8** | 130.7 | 121.5 | 131.0$_{\pm 0.46}$ |
| | Large | 58.6 | 135.4 | 133.4 | 123.0 | **151.9**$_{\pm 2.66}$ |
| **Single-task Average** | | 47.0 | 125.3 | 126.2 | 119.5 | **136.3** |
| Multi2d | U-Maze | 24.8 | 128.3 | 129.4 | 128.9 | **133.1**$_{\pm 0.41}$ |
| | Medium | 12.1 | 130.0 | **135.4** | 127.2 | 129.1$_{\pm 0.89}$ |
| | Large | 13.9 | 148.3 | 143.8 | 132.1 | **154.7**$_{\pm 2.79}$ |
| **Multi-task Average** | | 16.9 | 136.4 | 136.2 | 129.4 | **138.9** |

task settings. LoMAP effectively reduces manifold deviation (Section 4.1) and provides an additional performance boost, with Diffuser$^{\mathcal{P}}$ achieving the best results on 4 out of 6 tasks, showing especially strong improvements in Maze2D-Large, which features more complex obstacle maps.

**Locomotion.** We next evaluate LoMAP-incorporated planners on MuJoCo locomotion tasks (Fu et al., 2020), a standard benchmarks for assessing performance on heterogeneous, varying-quality datasets. Our comparison includes model-free algorithms (CQL (Kumar et al., 2020), IQL (Kostrikov et al., 2022)), model-based algorithms (MOPO (Yu et al., 2020), MOReL (Kidambi et al., 2020)), and sequence modeling approaches (Decision Transformer (DT) (Chen et al., 2021), Trajectory Transformer (TT) (Janner et al., 2021)). As baseline diffusion planners, we consider Diffuser (Janner et al., 2022), RGG (Lee et al., 2023b), TAT (Feng et al., 2024), and a conditional variant, Decision Diffuser (DD) (Ajay et al., 2023).

As shown in Table 2, incorporating LoMAP consistently boosts average returns of Diffuser across all tasks, with particularly strong gains in the Medium dataset, which poses a suboptimal and challenging distribution for learning both the diffusion planner and return estimator. Moreover, LoMAP-incorporated planners outperform other trajectory-refinement methods, highlighting the benefits of addressing manifold deviation during sampling.

### 4.3. Scaling to Hierarchical Planning in AntMaze

The AntMaze tasks (Fu et al., 2020) pose a substantial challenge due to high-dimensional state and action spaces, long-horizon navigation objectives, and sparse rewards. Generating entire trajectories often results in infeasible plans in these environments. A promising approach is to adopt a hierarchical scheme, wherein a high-level diffusion planner proposes subgoals and a low-level diffusion planner executes short-horizon trajectories to move the agent from one subgoal to the next.

Table 2: Performance comparison of Diffuser$^{\mathcal{P}}$ and various prior methods on MuJoCo locomotion tasks, reported as normalized average returns with corresponding standard errors over 50 planning seeds.

| Dataset | Environment | BC | CQL | IQL | DT | TT | MOPO | MOReL | DD | TAT | RGG | Diffuser | Diffuser$^{\mathcal{P}}$ |
|---|---|---|---|---|---|---|---|---|---|---|---|---|---|
| Med-Expert | HalfCheetah | 55.2 | 91.6 | 86.7 | 86.8 | 95.0 | 63.3 | 53.3 | 90.6 | 92.5 | 90.8 | 88.9 | $91.1_{\pm 0.23}$ |
| | Hopper | 52.5 | 105.4 | 91.5 | 107.6 | 110.0 | 23.7 | 108.7 | 111.8 | 109.4 | 109.6 | 103.3 | $110.6_{\pm 0.29}$ |
| | Walker2d | 107.5 | 108.8 | 109.6 | 108.1 | 101.9 | 44.6 | 95.6 | 108.8 | 108.8 | 107.8 | 106.9 | $109.2_{\pm 0.05}$ |
| Medium | HalfCheetah | 42.6 | 44.0 | 47.4 | 42.6 | 46.9 | 42.3 | 42.1 | 49.1 | 44.3 | 44.0 | 42.8 | $45.4_{\pm 0.13}$ |
| | Hopper | 52.9 | 58.5 | 66.3 | 67.6 | 61.1 | 28.0 | 95.4 | 79.3 | 82.6 | 82.5 | 74.3 | $93.7_{\pm 1.54}$ |
| | Walker2d | 75.3 | 72.5 | 78.3 | 74.0 | 79.0 | 17.8 | 77.8 | 82.5 | 81.0 | 81.7 | 79.6 | $79.9_{\pm 1.21}$ |
| Med-Replay | HalfCheetah | 36.6 | 45.5 | 44.2 | 36.6 | 41.9 | 53.1 | 40.2 | 39.3 | 39.2 | 41.0 | 37.7 | $39.1_{\pm 0.99}$ |
| | Hopper | 18.1 | 95.0 | 94.7 | 82.7 | 91.5 | 67.5 | 93.6 | 100 | 95.3 | 95.2 | 93.6 | $97.6_{\pm 0.58}$ |
| | Walker2d | 26.0 | 77.2 | 73.9 | 66.6 | 82.6 | 39.0 | 49.8 | 75 | 78.2 | 78.3 | 70.6 | $78.7_{\pm 2.2}$ |
| **Average** | | 51.9 | 77.6 | 77.0 | 74.7 | 78.9 | 42.1 | 72.9 | 81.8 | 81.3 | 81.2 | 77.5 | **82.8** |

Table 3: Performance comparison of Diffuser$^{\mathcal{P}}$, HD$^{\mathcal{P}}$, and prior approaches on AntMaze tasks, reported as normalized average returns with corresponding standard errors over 150 planning seeds.

| Dataset | Env | DD | RGG | Diffuser | Diffuser$^{\mathcal{P}}$ | HD | HD$^{\mathcal{P}}$ |
|---|---|---|---|---|---|---|---|
| Play | Medium | 8.0 | 17.3 | 6.7 | $40.7_{\pm 4.3}$ | 42.0 | $\mathbf{92.7}_{\pm 7.32}$ |
| | Large | 0.0 | 12.7 | 17.3 | $20.7_{\pm 3.8}$ | 54.7 | $\mathbf{74.0}_{\pm 6.2}$ |
| Diverse | Medium | 4.0 | 25.3 | 2.0 | $36.0_{\pm 3.7}$ | 78.7 | $\mathbf{98.0}_{\pm 6.1}$ |
| | Large | 0.0 | 17.3 | 27.3 | $39.3_{\pm 2.5}$ | 46.0 | $\mathbf{82.0}_{\pm 5.3}$ |
| **Average** | | 3.0 | 18.2 | 13.3 | 34.2 | 55.3 | **86.7** |

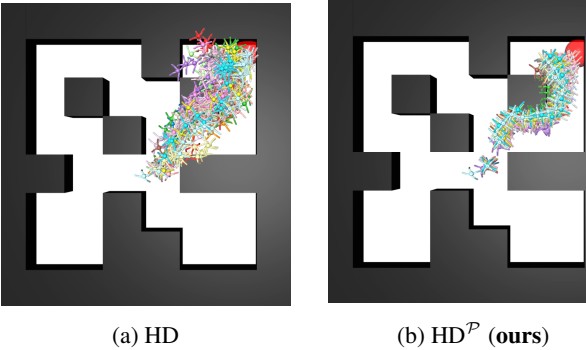

(a) HD      (b) HD$^{\mathcal{P}}$ (**ours**)

Figure 4: Visual comparison of generated plans on the AntMaze environment. The goal is marked by a red sphere. We plot 20 sampled plans in different colors. (a) shows plans generated by Hierarchical Diffuser (HD) (Chen et al., 2024), which often produces infeasible trajectories that pass through walls. (b) demonstrates the results of HD augmented with LoMAP, which respects the environment geometry and generates more reliable, feasible trajectories.

Building on this idea, we incorporate LoMAP into both Diffuser (Janner et al., 2022) and Hierarchical Diffuser (HD) (Chen et al., 2024), yielding Diffuser$^{\mathcal{P}}$ and HD$^{\mathcal{P}}$. In HD$^{\mathcal{P}}$, a high-level diffusion model augmented with LoMAP generates subgoals for each trajectory segment. Subsequently, a short-horizon diffusion model (Diffuser) translates these subgoals into lower-level actions. We compare Diffuser$^{\mathcal{P}}$ and HD$^{\mathcal{P}}$ against standard Diffuser (Janner et al., 2022), Hierarchical Diffuser (HD) (Chen et al., 2024), Restoration Gap Guidance (RGG) (Lee et al., 2023b), and Decision Diffuser (DD) (Ajay et al., 2023).

As shown in Table 3, Diffuser$^{\mathcal{P}}$ improves upon Diffuser across all AntMaze tasks, demonstrating ability of LoMAP to maintain manifold feasibility even in high-dimensional continuous control. Notably, HD$^{\mathcal{P}}$ achieves the best results on every variant of AntMaze, substantially outperforming the original HD. We attribute these gains largely to the correction of manifold deviation during high-level planning by LoMAP. In standard HD, subgoals generated by the high-level planner can sometimes lie off-manifold, forcing the low-level planner to produce infeasible trajectories. As illustrated in Figure 4, these subgoals frequently pass through maze walls, leading to invalid paths. By applying LoMAP to refine them, HD$^{\mathcal{P}}$ ensures that each proposed subgoal is more feasible for the low-level planner, thereby boosting overall success rates. These improvements are especially pronounced in larger mazes, where longer horizons and intricate navigation paths make adherence to a valid manifold particularly critical. Consequently, LoMAP serves as a plug-and-play component that boosts planning performance even in multi-level hierarchical settings.

### 4.4. Generating Minority Sample

The ability to generate minority data can be critical in real-world scenarios where increasing the diversity of rare-condition examples can improve predictive performance. However, any minority samples must still align with the true data distribution rather than represent artifacts. To explore whether our method can facilitate the generation of *feasible* minority samples in low-density regions, we adopt minority guidance (Um et al., 2024), which provides additional guidance toward low-density regions.

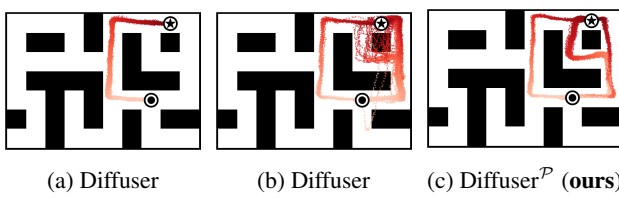

| (a) Diffuser | (b) Diffuser | (c) Diffuser$^{\mathcal{P}}$ (**ours**) |
| | +minority guidance | +minority guidance |

Figure 5: Sampling from low-density regions in Maze2D using minority guidance (Um et al., 2024), given a specified start ⊚ and goal ✪ condition.

As shown in Figure 5, Diffuser alone sometimes fails to capture alternative feasible paths, leading to poor coverage despite viable shortcuts. While minority guidance improves coverage, it also tends to introduce infeasible trajectories. In contrast, LoMAP mitigates this issue by refining these trajectories and ensuring they remain on the valid manifold. Consequently, combining LoMAP with minority guidance can help uncover feasible yet unexplored solutions that might otherwise remain inaccessible to standard diffusion planners. Investigating how this approach can further enhance planning is a promising direction for future work.

## 5. Related Work

**Diffusion Planners in Offline Reinforcement Learning**
Diffusion probabilistic models (Sohl-Dickstein et al., 2015; Ho et al., 2020) have recently gained prominence in reinforcement learning (RL), particularly in the offline setting. By iteratively denoising samples from noise, these models learn the gradient of the data distribution (Song & Ermon, 2019), bridging connections to score matching (Hyvärinen & Dayan, 2005) and energy-based models (EBMs) (Du & Mordatch, 2019; Grathwohl et al., 2020). Their expressive power in modeling complex, high-dimensional data has led to applications as planners (Janner et al., 2022; Ajay et al., 2023), policies (Wang et al., 2023), and data synthesizers (Lu et al., 2023a; Wang et al., 2025).

Diffuser (Janner et al., 2022) pioneered the use of diffusion models for planning by generating entire trajectories, demonstrating notable flexibility in long-horizon tasks. Concretely, an unconditional diffusion model is trained on offline trajectories and paired with a separate network that estimates returns; this network then guides trajectory samples toward high-return regions during inference (Dhariwal & Nichol, 2021). Extending this framework, Decision Diffuser (Ajay et al., 2023) applies *classifier-free guidance*, conditioning the diffusion model directly on reward or constraint signals and thereby removing the need for a separately trained reward function. Meanwhile, AdaptDiffuser (Liang et al., 2023) progressively fine-tunes the diffusion model with high-quality synthetic data, improving generalization in goal-conditioned tasks. Beyond these efforts, diffusion models have also been employed in hierarchical

planning (Li et al., 2023; Chen et al., 2024), multi-task RL (He et al., 2023; Ni et al., 2023), and multi-agent settings (Zhu et al., 2024).

Despite these advances, diffusion planners remain susceptible to *stochastic failures*, occasionally producing trajectories that deviate from the feasible manifold. Although some works mitigate this issue by refining trajectories post hoc (Lee et al., 2023b; Feng et al., 2024), a robust, training-free approach to consistently maintain manifold adherence throughout the sampling process has yet to be established.

**Projections in Diffusion Models** Several works in the image-generation domain have introduced projection techniques to mitigate off-manifold updates during diffusion sampling. For instance, MCG (Chung et al., 2022) projects measurement gradients onto the data manifold in inverse problems, guided by Tweedie's formula. DSG (Yang et al., 2024) replaces the random Gaussian step with a deterministic update constrained to a hypersphere. This avoids deviating from the intermediate diffusion manifold, allowing for substantially larger guidance steps. Meanwhile, MPGD (He et al., 2024) employs a pre-trained autoencoder to learn the data manifold and projects the sample onto the tangent space of the clean data manifold via a pre-trained autoencoder. However, the performance of MPGD is heavily depends on the expressive power of the autoencoder. As a result, it is difficult to deploy MPGD in diverse offline RL tasks, particularly where pre-trained autoencoders are unavailable. In contrast, our proposed method is entirely *training-free*, projecting samples onto *both* the clean and intermediate diffusion manifolds using only local approximations from the offline dataset.

## 6. Conclusion

In this work, we investigated the manifold deviation issue that arises in diffusion-based trajectory planning, where inaccurate guidance causes sampled trajectories to deviate from the feasible data manifold. To address this, we introduced *Local Manifold Approximation and Projection* (LoMAP), a training-free method that employs local, low-rank projections to constrain each reverse diffusion step to the underlying data manifold. By ensuring that intermediate samples remain close to this manifold, LoMAP substantially reduces the risk of generating infeasible or low-quality trajectories. Empirical results on various offline RL benchmarks demonstrate the effectiveness of our approach. Additionally, LoMAP can be incorporated into hierarchical diffusion planning for more challenging tasks such as AntMaze. Overall, our results establish LoMAP as an easily integrable component for diffusion-based planners, empowering them to consistently remain on the data manifold and thereby providing safer and more robust long-horizon trajectories.

## Acknowledgements

This work was supported by Institute of Information & communications Technology Planning & Evaluation (IITP) grant funded by the Korea government (MSIT) (No.2019-0-00075, Artificial Intelligence Graduate School Program (KAIST); No. 2022-0-00984, Development of Artificial Intelligence Technology for Personalized Plug-and-Play Explanation and Verification of Explanation; No. RS-2024-00457882, AI Research Hub Project; No. RS-2024-00509258, AI Guardians: Development of Robust, Controllable, and Unbiased Trustworthy AI Technology).

## Impact Statement

This paper advances the field of Machine Learning through a new method for diffusion-based trajectory planning. We do not identify any direct negative societal impacts that must be specifically highlighted. However, we encourage practitioners to apply this work responsibly and evaluate real-world safety implications before deployment.

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

# A. Proofs

**Proposition A.1.** *(Dimensional scaling of guidance gap.) Suppose $\mathcal{J}(\boldsymbol{\tau}^0)$ is not constant. Given the* true *guidance*

$$\nabla_{\boldsymbol{\tau}^i}\,\mathcal{J}_t(\boldsymbol{\tau}^i) = \frac{\mathbb{E}_{q(\boldsymbol{\tau}^0|\boldsymbol{\tau}^i)}\big[e^{\mathcal{J}(\boldsymbol{\tau}^0)}\,\nabla_{\boldsymbol{\tau}^i}\log q(\boldsymbol{\tau}^0|\boldsymbol{\tau}^i)\big]}{\mathbb{E}_{q(\boldsymbol{\tau}^0|\boldsymbol{\tau}^i)}\big[e^{\mathcal{J}(\boldsymbol{\tau}^0)}\big]},$$

*and the* MSE-based *guidance*

$$\nabla_{\boldsymbol{\tau}^i}\,\mathcal{J}_\phi^{\mathrm{MSE}}(\boldsymbol{\tau}^i) = \mathbb{E}_{q(\boldsymbol{\tau}^0|\boldsymbol{\tau}^i)}\big[\mathcal{J}(\boldsymbol{\tau}^0)\,\nabla_{\boldsymbol{\tau}^i}\log q(\boldsymbol{\tau}^0|\boldsymbol{\tau}^i)\big],$$

*there exists a choice of $\boldsymbol{\tau}^i$ such that*

$$\Big\|\nabla_{\boldsymbol{\tau}^i}\,\mathcal{J}_t(\boldsymbol{\tau}^i) - \nabla_{\boldsymbol{\tau}^i}\,\mathcal{J}_\phi^{\mathrm{MSE}}(\boldsymbol{\tau}^i)\Big\|_2 \geq \frac{c}{\sqrt{1-\alpha_i}}\sqrt{d},$$

*for some constant $c > 0$ that does not depend on $d$.*

*Proof.* By the forward process at in Eq. (4),

$$\boldsymbol{\tau}^i = \sqrt{\alpha_i}\,\boldsymbol{\tau}^0 + \sqrt{1-\alpha_i}\,\boldsymbol{\epsilon}, \quad \boldsymbol{\epsilon} \sim \mathcal{N}(\mathbf{0},\mathbf{I}),$$

We have,

$$q(\boldsymbol{\tau}^0|\boldsymbol{\tau}^i) = \mathcal{N}\Big(\frac{\boldsymbol{\tau}^i}{\sqrt{\alpha_i}}, \frac{1-\alpha_i}{\alpha_i}\,\mathbf{I}_d\Big), \quad \nabla_{\boldsymbol{\tau}^i}\log q(\boldsymbol{\tau}^0|\boldsymbol{\tau}^i) = -\frac{1}{\sqrt{1-\alpha_i}}\,\boldsymbol{\epsilon}.$$

Let us abbreviate the distribution $\mu(\boldsymbol{\tau}^0) := q(\boldsymbol{\tau}^0|\boldsymbol{\tau}^i)$. Then

$$\nabla_{\boldsymbol{\tau}^i}\,\mathcal{J}_t(\boldsymbol{\tau}^i) = \frac{\mathbb{E}_\mu\big[e^{\mathcal{J}(\boldsymbol{\tau}^0)}\,\nabla_{\boldsymbol{\tau}^i}\log\mu(\boldsymbol{\tau}^0)\big]}{\mathbb{E}_\mu\big[e^{\mathcal{J}(\boldsymbol{\tau}^0)}\big]}, \quad \nabla_{\boldsymbol{\tau}^i}\,\mathcal{J}_\phi^{\mathrm{MSE}}(\boldsymbol{\tau}^i) = \mathbb{E}_\mu\big[\mathcal{J}(\boldsymbol{\tau}^0)\,\nabla_{\boldsymbol{\tau}^i}\log\mu(\boldsymbol{\tau}^0)\big].$$

Subtracting these yields

$$\nabla_{\boldsymbol{\tau}^i}\,\mathcal{J}_t(\boldsymbol{\tau}^i) - \nabla_{\boldsymbol{\tau}^i}\,\mathcal{J}_\phi^{\mathrm{MSE}}(\boldsymbol{\tau}^i) = \mathbb{E}_\mu\Big[\Big(\frac{e^{\mathcal{J}(\boldsymbol{\tau}^0)}}{\mathbb{E}_\mu\big[e^{\mathcal{J}(\boldsymbol{\tau}^0)}\big]} - \mathcal{J}(\boldsymbol{\tau}^0)\Big)\nabla_{\boldsymbol{\tau}^i}\log\mu(\boldsymbol{\tau}^0)\Big].$$

Since $\nabla_{\boldsymbol{\tau}^i}\log\mu(\boldsymbol{\tau}^0) = -\frac{1}{\sqrt{1-\alpha_i}}\,\boldsymbol{\epsilon}$, we obtain

$$\nabla_{\boldsymbol{\tau}^i}\,\mathcal{J}_t(\boldsymbol{\tau}^i) - \nabla_{\boldsymbol{\tau}^i}\,\mathcal{J}_\phi^{\mathrm{MSE}}(\boldsymbol{\tau}^i) = -\frac{1}{\sqrt{1-\alpha_i}}\,\mathbb{E}\big[\delta(\boldsymbol{\tau}^0)\,\boldsymbol{\epsilon}\big].$$

By Jensen's inequality, $\delta(\boldsymbol{\tau}^0)$ has positive mean whenever $\mathcal{J}$ is not constant. Furthermore, as $d$ grows, we have $\|\boldsymbol{\epsilon}\|_2$ on the order of $\sqrt{d}$. One can then choose a $\boldsymbol{\tau}^i$ so that $\delta(\boldsymbol{\tau}^0)$ remains well-aligned with $\boldsymbol{\epsilon}$, giving

$$\Delta_{\mathrm{guidance}}(\boldsymbol{\tau}^i) = \frac{1}{\sqrt{1-\alpha_i}}\big\|\mathbb{E}\big[\delta(\boldsymbol{\tau}^0)\,\boldsymbol{\epsilon}\big]\big\|_2 \geq \frac{c}{\sqrt{1-\alpha_i}}\sqrt{d}.$$

for some constant $c$. Thus we can complete the proof.

$\square$

# B. Limitations

While LoMAP provides a simple yet effective approach for mitigating manifold deviations, it exhibits certain limitations. First, the current implementation uses cosine distance for manifold approximation, which may not be optimal in very high-dimensional state spaces, such as pixel-based observations. Developing more robust manifold approximation techniques suitable for complex, high-dimensional environments remains an important direction for future research. For instance, combining LoMAP with latent trajectory embeddings (Co-Reyes et al., 2018) could be a promising approach. Second, our

method inherently encourages sampled trajectories to stay close to the offline data manifold, which may restrict exploration of novel behaviors. While our primary focus in this work is ensuring safe and reliable trajectory generation—particularly beneficial for safety-critical offline RL applications—addressing this exploration limitation remains crucial. Integrating LoMAP with complementary methods such as trajectory stitching or data augmentation (Ziebart et al., 2008; Li et al., 2024; Lee et al., 2024a; Yang & Wang, 2025), which generate diverse synthetic trajectories, could alleviate this issue and is an interesting area for future study. Furthermore, exploring how LoMAP could be effectively extended to challenging benchmarks that explicitly require stitching and long-horizon reasoning, such as OGBench (Park et al., 2025), represents an intriguing future research direction.

## C. Extended Related Work

Beyond hierarchical structures (Li et al., 2023; Chen et al., 2024), multi-agent setups (Zhu et al., 2024), and post-hoc trajectory refinement methods (Lee et al., 2023b; Feng et al., 2024), recent diffusion planners have explored integrating tree search methods (Yoon et al., 2025), refining trajectory sampling techniques (Dong et al., 2024a), examining critical design choices to improve robustness (Lu et al., 2025), composing short segments into long-horizon trajectories at inference time (Mishra et al., 2023; Luo et al., 2025), efficient latent diffusion planning (Li, 2024), and inference-time guided generation (Wang et al., 2024; Lee et al., 2024b; Hao et al., 2024).

While diffusion models have achieved impressive performance on various generative tasks, effectively steering them toward specific objectives remains challenging. Broadly, existing methods for aligning diffusion models can be categorized into two groups: fine-tuning methods and guidance-based methods. Fine-tuning methods, such as reinforcement learning-based tuning (Fan et al., 2023) or direct gradient optimization (Clark et al., 2024; Prabhudesai et al., 2024), directly update model parameters to maximize target objectives. Despite their effectiveness, these methods tend to excessively focus on reward optimization, often compromising the diversity and fidelity of generated outputs (Clark et al., 2024). Conversely, guidance-based methods offer a simpler inference-time alternative that preserves the pretrained model distribution. Among these, classifier guidance (Dhariwal & Nichol, 2021) involves training an auxiliary classifier to guide the sampling process toward target conditions, but the additional training overhead can be costly. Recent training-free guidance approaches circumvent this by directly utilizing pretrained classifiers or reward predictors via approximate inference (Chung et al., 2023; Song et al., 2023; He et al., 2024). In particular, these methods commonly rely on Tweedie-based denoising (Robbins, 1992), which provides predictions of clean data given noisy samples. However, inaccuracies inherent to Tweedie's approximation limit its effectiveness, especially in accurately aligning diffusion samples with target objectives. Sequential Monte Carlo (SMC)-based approaches (Wu et al., 2023; Cardoso et al., 2024) address inaccuracies in guidance through principled probabilistic inference. Although these methods provide asymptotic exactness, their practical efficiency under limited sampling budgets remains a significant challenge.

## D. Additional Results

**Realism score evaluation.** To further validate the effectiveness of LoMAP, we compute the Realism Score (Kynkäänniemi et al., 2019), which measures how closely generated trajectories lie to the true manifold defined by the offline dataset. Specifically, we approximate the true manifold using $k$-nearest neighbor (k-NN) hyperspheres constructed from 20,000 offline trajectories, and evaluate the average realism score over 100,000 sampled trajectories. As shown in Table 4, applying LoMAP consistently yields higher realism scores compared to diffusion sampling without LoMAP, demonstrating that LoMAP effectively produces trajectories closer to the true data manifold.

Table 4: Comparison of Realism Scores on Maze2D tasks. Higher realism scores indicate samples closer to the true data manifold.

| Environment | Diffuser | Diffuser$^{\mathcal{P}}$ |
|---|---|---|
| Maze2D U-Maze | 1.23 | **1.30** |
| Maze2D Medium | 1.40 | **1.56** |
| Maze2D Large | 1.36 | **1.47** |

**Dynamic consistency evaluation.** Additionally, we assess trajectory feasibility for MuJoCo locomotion tasks using the dynamic mean squared error (Dynamic MSE), defined as:

$$\text{Dynamic MSE} = \|f^*(\boldsymbol{s}, \boldsymbol{a}) - \boldsymbol{s}'\|_2^2,$$

where $f^*$ represents the true environment dynamics. As shown in Table 5, LoMAP consistently achieves lower Dynamic MSE compared to diffusion sampling without LoMAP, clearly indicating improved adherence to the true dynamics of the

environment.

Table 5: Dynamic MSE comparison on MuJoCo locomotion tasks. Lower Dynamic MSE indicates better adherence to true environment dynamics.

| Environment | Diffuser | Diffuser$^{\mathcal{P}}$ |
|---|---|---|
| halfcheetah-medium-expert | 0.363 | **0.295** |
| hopper-medium-expert | 0.027 | **0.020** |
| walker2d-medium-expert | 0.391 | **0.293** |
| halfcheetah-medium | 0.352 | **0.285** |
| hopper-medium | 0.024 | **0.021** |
| walker2d-medium | 0.395 | **0.293** |
| halfcheetah-medium-replay | 0.710 | **0.555** |
| hopper-medium-replay | 0.049 | **0.045** |
| walker2d-medium-replay | 0.829 | **0.506** |

**Additional comparison with inference-time guidance methods.** We further provide comparative evaluations against recent inference-time guidance methods, including stochastic sampling (Wang et al., 2024), constrained gradient guidance (Lee et al., 2024b), and inpainting optimization (Hao et al., 2024).

For stochastic sampling (Wang et al., 2024), we adapted goal-conditioning via MCMC sampling, tuning the number of sampling steps $\{2, 4, 6, 8\}$. To implement constrained gradient guidance (Lee et al., 2024b), we approximated maze walls as multiple spherical constraints following Shaoul et al. (2024), defining a sphere-based cost:

$$J_c(\boldsymbol{\tau}) = \sum_{m=1}^{M} \sum_{t=1}^{H} \max\left(r - \mathrm{dist}(\boldsymbol{\tau}_t, \boldsymbol{p}_m), 0\right),$$

where $H$ is the planning horizon, $\boldsymbol{p}_m$ the center of sphere constraints, and $r$ their radius. We tuned the guidance scale within the range $\{0.001, 0.01, 0.05, 0.1\}$. To compare with inpainting optimization (Hao et al., 2024), we emulated a vision-language model (VLM)-based keyframe generation by training a high-level policy using Hierarchical Implicit Q-Learning (HIQL) (Park et al., 2023). The policy generated optimal subgoal sequences (keyframes), with $k = 25$ steps, aligning with the official implementation provided by ogbench (Park et al., 2025).

Table 6 presents artifact ratio comparisons. LoMAP consistently achieves the lowest artifact ratio, significantly outperforming all inference-time guidance baselines. Even the constrained gradient approach (Lee et al., 2024b), despite explicitly modeling maze walls, performed worse, likely due to gradient-based projections struggling with nonconvex constraints. Both stochastic sampling (Wang et al., 2024) and inpainting optimization (Hao et al., 2024) improved over Diffuser but still exhibited higher artifact ratios than LoMAP.

Table 6: Artifact ratio comparison with inference-time guidance methods in Maze2D-Large. Lower values indicate fewer infeasible trajectories.

| # of Plans | Diffuser$^{\mathcal{P}}$ (LoMAP, ours) | Diffuser | (Wang et al., 2024) | (Lee et al., 2024b) | (Hao et al., 2024) |
|---|---|---|---|---|---|
| 10 | **0.35** | 0.50 | 0.42 | 0.49 | 0.43 |
| 20 | **0.35** | 0.62 | 0.44 | 0.54 | 0.46 |
| 30 | **0.38** | 0.66 | 0.47 | 0.61 | 0.49 |

**Visual comparisons.** We provide additional rollout visualizations. Figure 6 depicts rollouts executed by Diffuser (Janner et al., 2022) and our Diffuser$^{\mathcal{P}}$ on the Hopper-Medium dataset, demonstrating effectiveness even in suboptimal and challenging data distribution. Meanwhile, Figure 7 offers a side-by-side comparison of Diffuser and HD$^{\mathcal{P}}$ on AntMaze-Large-Diverse. We observe that, while standard Diffuser frequently produces trajectories that collide with maze walls or fail to reach the goal, our hierarchical extension with LoMAP (i.e., HD$^{\mathcal{P}}$) maintains more coherent routes and significantly

increases the likelihood of reaching the target (marked by the red sphere). In both examples, projecting intermediate diffusion steps onto a locally approximated manifold substantially mitigates stochastic failures, highlighting the effectiveness of our approach for long-horizon, high-dimensional control tasks.

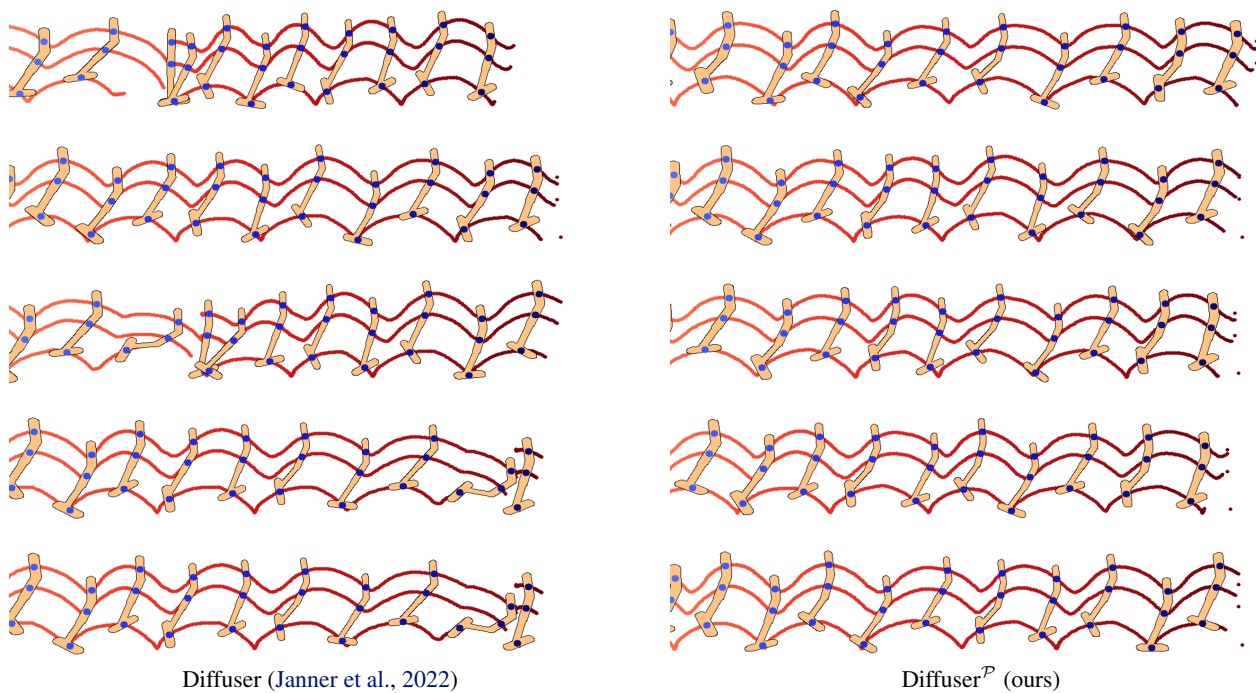

Diffuser (Janner et al., 2022)          Diffuser$^{\mathcal{P}}$ (ours)

Figure 6: Visual comparison of rollout trajectories from Diffuser and Diffuser$^{\mathcal{P}}$ in the Hopper-Medium task.

# E. Experimental Details

### E.1. Environments

**Maze2D.** Maze2D environments (Fu et al., 2020) require an agent to undertake long-horizon navigation, moving toward a distant goal location. The agent accrues no reward unless it successfully reaches the goal, at which point it receives a reward of 1. The three available layouts—U-Maze, Medium, and Large—vary in complexity. Further, Maze2D features two distinct tasks: a single-task variant with a fixed goal and a multi-task option (*Multi2D*), which randomizes the goal location at the beginning of each episode. A summary of key details can be found in Table 7.

Table 7: Environment details for Maze2D experiments.

|  | Maze2D-Large | Maze2D-Medium | Maze2D-UMaze |
|---|---|---|---|
| State space $\mathcal{S}$ | $\mathbb{R}^4$ | $\mathbb{R}^4$ | $\mathbb{R}^4$ |
| Action space $\mathcal{A}$ | $\mathbb{R}^2$ | $\mathbb{R}^2$ | $\mathbb{R}^2$ |
| Goal space $\mathcal{G}$ | $\mathbb{R}^2$ | $\mathbb{R}^2$ | $\mathbb{R}^2$ |
| Episode length | 800 | 600 | 300 |

**Locomotion.** Gym-MuJoCo locomotion tasks (Fu et al., 2020) serve as widely recognized benchmarks for assessing algorithm performance on heterogeneous datasets of varying quality. The `Medium` dataset consists of one million samples gathered from an SAC (Haarnoja et al., 2018) agent trained to roughly one-third of expert-level performance. The `Medium-Replay` dataset contains all experiences accumulated throughout the SAC training process up to that same performance threshold. Finally, the `Medium-Expert` dataset is created by combining expert demonstrations and suboptimal

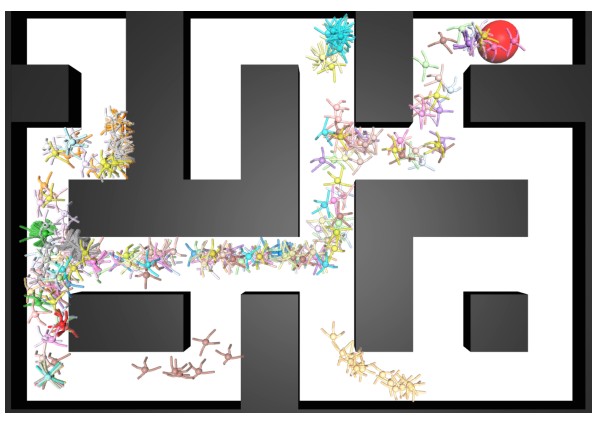
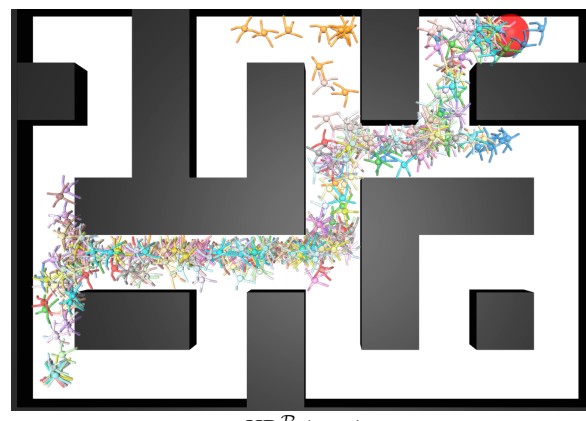

Diffuser (Janner et al., 2022)  HD$^{\mathcal{P}}$ (**ours**)

Figure 7: Visual comparison of rollout trajectories from Diffuser and HD$^{\mathcal{P}}$ in the AntMaze-Large-Diverse task. We plot each trajectory at 50-step intervals for clarity. The goal is marked by a red sphere, and 20 rollout trajectories are shown in different colors.

data in equal amounts. Details on the state/action spaces and episode lengths can be found in Table 8.

Table 8: Environment details for Locomotion experiments.

|  | **Hopper-*** | **Walker2d-*** | **Halfcheetah-*** |
|---|---|---|---|
| State space $\mathcal{S}$ | $\mathbb{R}^{11}$ | $\mathbb{R}^{17}$ | $\mathbb{R}^{17}$ |
| Action space $\mathcal{A}$ | $\mathbb{R}^{3}$ | $\mathbb{R}^{6}$ | $\mathbb{R}^{6}$ |
| Episode length | 1000 | 1000 | 1000 |

**AntMaze.** AntMaze tasks (Fu et al., 2020) involve guiding an 8-DoF *Ant* robot through intricate maze layouts using MuJoCo for physics simulation. The environment uses a sparse reward structure, granting a reward only upon reaching the designated goal, thus posing a challenging long-horizon navigation problem. Further difficulty arises from the offline dataset, which contains numerous trajectory segments that do not succeed in reaching the goal. We measure performance by the success rate of the agent reaching the endpoint. A summary of the key environment details is provided in Table 9.

Table 9: Environment details for AntMaze experiments.

|  | **AntMaze-*** |
|---|---|
| State space $\mathcal{S}$ | $\mathbb{R}^{29}$ |
| Action space $\mathcal{A}$ | $\mathbb{R}^{8}$ |
| Episode length | 1000 |

### E.2. Implementation Details

Below, we summarize the key implementation details and hyperparameters used throughout our experiments:

- **Network architecture.** We build on the Diffuser framework (Janner et al., 2022), employing a temporal U-Net with repeated convolutional residual blocks to parameterize $\epsilon_\theta$.

- **Planning horizons.** For Maze2D and Multi2D tasks, the planning horizon is 128 in U-Maze, 256 in Medium, and 256 in Large. For MuJoCo locomotion tasks, the horizon is 32, and for AntMaze it is 64.

- **Diffusion steps.** We use 256 steps for the diffusion process in Maze2D//Multi2D Large and Medium, 128 in Maze2D//Multi2D U-Maze, and 20 in other environments.

- **Guidance scales.** For AntMaze tasks, we select the guidance scale $\omega$ from the set $\{5.0, 3.0, 1.0, 0.1, 0.01, 0.001\}$. In MuJoCo locomotion tasks, we select $\omega$ from $\{0.3, 0.2, 0.1, 0.01, 0.001, 0.0001\}$ during planning.

- **Local manifold approximation.** We tune the number of neighbors $k \in \{5, 10, 20\}$ in our local manifold approximation procedure.

- **Hierarchical Diffuser in AntMaze.** For the high-level and low-level planners, we follow Chen et al. (2024) and train each component separately using trajectory segments randomly sampled from the D4RL offline dataset. Specifically, the high-level planner generates state-space trajectories with a planning horizon of 226 and temporal jumps of 15. During execution, the corresponding actions are inferred through a learned inverse dynamics model (Ajay et al., 2023).

## F. Practical Implementation

**Manifold approximation.** A straightforward $k$-nearest-neighbor retrieval from the entire offline dataset at each diffusion step can be prohibitively expensive. To mitigate this cost, we employ an *inverted file* (IVF) index from the Faiss library (Douze et al., 2024), which partitions the dataset into a set of coarse centroids and restricts each query to only a few relevant clusters.

Concretely, IVF uses $k$-means to learn $n_{\text{list}}$ centroids $\{\mathbf{c}_1, \dots, \mathbf{c}_{n_{\text{list}}}\}$ across the dataset of dimension $d$. Each data point $\mathbf{x}$ is mapped to its nearest centroid, forming an inverted list. A query vector $\mathbf{q}$ is matched to its closest centroids, after which the search proceeds solely within the corresponding clusters. This design significantly reduces the number of distance computations relative to an exhaustive linear scan. Although coarse clustering can introduce minor inaccuracies, we have found this approach to be effective in our experiments.

In LoMAP, the IVF-based approximate neighbor search operates on a denoised sample $\hat{\boldsymbol{\tau}}^{0|i}$, retrieving up to $k$ neighbors from the offline dataset. We then forward-diffuse these neighbors to timestep $i$ and perform a rank-$r$ principal-component analysis to approximate the local manifold of feasible trajectories. Projecting $\boldsymbol{\tau}^i$ onto this manifold helps correct off-manifold drift caused by inexact guidance. Crucially, limiting the search to relevant clusters enables LoMAP to handle large datasets and high-dimensional state-action spaces, making it practical for long-horizon offline reinforcement-learning tasks.

**Manifold projection.** We observed that applying manifold projection selectively, rather than uniformly across all diffusion steps, can significantly reduce computational costs and even enhance overall performance. Specifically, we found that projection is particularly beneficial when applied during intermediate to later stages of the reverse diffusion process. We hypothesize two main reasons for this phenomenon: first, Tweedie-based denoisers inherently exhibit biases toward majority or high-density features, as noted by Um et al. (2024), making early stage projections less impactful. Second, the discrepancy between the learned reverse transitions of the diffusion model and the true data distribution becomes most pronounced at intermediate diffusion steps, consistent with observations reported in prior studies (Na et al., 2024). Consequently, concentrating projection efforts on these critical stages not only improves computational efficiency but also effectively mitigates manifold deviation, resulting in improved sampling quality.

## G. Baseline Performance Sources

### G.1. Maze2D

The reported IQL scores come from Table 1 in Janner et al. (2022), RGG scores from Table 2 in Lee et al. (2023b), and TAT scores from Table 2 in Feng et al. (2024).

### G.2. Locomotion

We obtain scores for BC, CQL, and IQL from Table 1 in Kostrikov et al. (2022); DT from Table 2 in Chen et al. (2021); TT from Table 1 in Janner et al. (2021); MOPO from Table 1 in Yu et al. (2020); MOReL from Table 2 in Kidambi et al. (2020); and Diffuser from Table 2 in Janner et al. (2022). Scores for RGG and TAT are drawn from Table 3 in Feng et al. (2024), while DD scores come from Table 1 in Ajay et al. (2023).

### G.3. AntMaze

Scores for DD in the AntMaze domain are taken from Table 1 in Dong et al. (2024b).

