# OpenReview forum: "Local Manifold Approximation and Projection for Manifold-Aware Diffusion Planning"
_ICML.cc/2025/Conference — ICML 2025 poster_

### Official Review · Reviewer_bsde · 2025-03-13

**Overall Recommendation:** 3

**Summary:**

The authors introduce formally and investigate mathematically the manifold deviation issue due to approximate guidance in the context of trajectory-planning (for reward maximization) via diffusion models. They provide a lower bound on this error, and to address this issue they introduce LoMAP, a training-free method that performs projections sequentially by leveraging offline data along the reverse sampling process to constrain the generated samples to the underlying data manifold. Ultimately, they show extensive experimental validation for the proposed method.

## update after rebuttal
After reading the authors rebuttal I decided to keep my original score as the content of the rebuttal did not alter significantly my beliefs regarding positive aspects as well as (inherent) limitations of this work.

**Claims And Evidence:**

1 - The claim "The current sample is then projected onto this subspace, thereby remaining on the manifold of valid behaviors." at line 69 seems not precise. Diffusion models, as well of offline data, capture an implicit notion of validity only approximately. As a consequence these models can generate data points that are invalid, and the procedure presented within the paper, although useful to improve the chances of generating valid points, does not seem to imply that the generated points are valid. In order to achieve this, the diffusion model would need to interact with an available validity checker or certain assumptions would have to be made about the data space.

2 - The claim "Consequently, combining LoMAP with minority guidance can help uncover feasible yet unexplored solutions that might otherwise remain inaccessible to standard diffusion planners." at line 403 seems not precise. To the best of my knowledge, training a diffusion model consists in learning a function approximator for the score function (i.e., a score network) that can induce a final marginal density (on the data space) which modes (i.e., significantly positive density) in regions where no data are present. It seems to me that the proposed method would instead prevent the diffusion model from sampling these modes, while it would let the model sample low-density modes where enough offline data are present. From the experiments presented within Sec. 4.4 this distinction is not clear and therefore the claim above seems not supported, especially the 'unexplored solutions' part.

**Essential References Not Discussed:**

I am not aware of important references not mentioned.

**Experimental Designs Or Analyses:**

Checked exp. in Sec. 4.1 and seems convincing to me. Regarding exp. in Sec. 4.4 I mentioned a doubt within point (2) of the Claims and Evidence section.

**Methods And Evaluation Criteria:**

Yes.

**Other Comments Or Suggestions:**

- what is $n$ in line 161 defined?

**Other Strengths And Weaknesses:**

Strengths:
1. The paper tackles a timely open problem and presents a formally clear formulation.
2. Proposes a solution that seems of practical relevance (i.e., easy to use) and well-performing on relevant problems.
3. The paper is well written, clear, and easy to read.

Weaknesses:
1. My main concern regarding the paper regards the risk of ending up obtaining a sampler excessively regularized from offline data (i.e., overfitting to offline data) as explained within my point (2) of Claims And Evidence section.  This might be problematic towards using the introduced machinery in RL problems, where the goal is often to discover new strategies not contained within offline data.

**Questions For Authors:**

- Did I misinterpret something within the weakness point that I mentioned ?
- In line 225 the paper claims that Prop. 3.2 indicates that the manifold dev. problem is stronger for long-horizon tasks. How should I infer this from the mathematical result within the proposition?

**Relation To Broader Scientific Literature:**

- The paper tackles an important problem that arises in the context of trajectory-based planning, a very timely research topic, tackled via guidance. To the best of my knowledge, the paper properly mentions related works and references and makes a convincing positive comparison with works within this literature stream.

- The problem of reward-guided sampling can be solved in a multitude of ways beyond guidance. Examples include using RL[1] or control-theoretic[2] formulations to fine-tuning a diffusion model, or inference-time techniques beyond guidance [3]. It is not clear how the ideas presented in this work extend to these settings, as the current formulation seems quite specific to guidance.

[1] Understanding Reinforcement Learning-Based Fine-Tuning of Diffusion Models: A Tutorial and Review, Uehara et al.
[2] Adjoint Matching: Fine-tuning Flow and Diffusion Generative Models with Memoryless Stochastic Optimal Control, Domingo-Enrich et al.
[3] Inference-Time Alignment in Diffusion Models with Reward-Guided Generation: Tutorial and Review, Uehara et al.

**Theoretical Claims:**

Checked the derivations in the main paper and seem correct.

---

> ### Author Rebuttal · Authors · 2025-03-30
>
> Thank you for the detailed review and constructive feedback on our work. We especially appreciate the reviewer pointing out the clarity issue regarding our claims. Please find our detailed response below.
>
> - **“The claim "The current sample is then projected onto this subspace, thereby remaining on the manifold of valid behaviors." at line 69 seems not precise.”**
>
> We fully agree with the reviewer that our LoMAP method does not necessarily guarantee the validity of behaviors generated by diffusion models. Accordingly, we will **tone down** this claim to avoid ambiguity and overstatement. Specifically, we will revise:
>
> > "thereby remaining on the manifold of valid behaviors."
> >
>
> to a more accurate and precise description:
>
> > "thereby significantly reducing off-manifold deviations and improving the likelihood of generating valid behaviors."
> >
>
> - **“The claim "Consequently, combining LoMAP with minority guidance can help uncover feasible yet unexplored solutions that might otherwise remain inaccessible to standard diffusion planners." at line 403 seems not precise.”**
>
> We apologize for the confusion caused by this statement. Our original intention was to highlight that standard diffusion planners typically require a large number of samples to generate trajectories located in low-density regions of the data distribution, thus limiting exploration within a restricted sampling budget.
>
> We aimed to clarify that one way to overcome this limitation—though not our contribution—is through minority guidance [1], which explicitly guides sampling towards low-density regions. Our contribution, LoMAP, further enhances this process by ensuring that trajectories sampled under minority guidance remain feasible and closer to the data manifold, allowing efficient generation of trajectories from low-density regions without extensive sampling.
>
> We appreciate the reviewer highlighting this ambiguity and will clarify this point carefully in the final version of the paper.
>
> - **“what is $n$ in line 161 defined?”**
>
> We apologize for this confusion. This was a typo. To clarify, the original sentence:
>
> > "… $\boldsymbol{\tau}^i$ is inherently concentrated around a ($n-d$) dimensional manifold $\mathcal{M}_i$."
> >
>
> should be corrected to the following description:
>
> > "… $\boldsymbol{\tau}^i$ is inherently concentrated around a ($d-k$)-dimensional manifold $\mathcal{M}_i$."
> >
>
> Thank you for pointing out this typo.
>
> - **“In line 225 the paper claims that Prop. 3.2 indicates that the manifold dev. problem is stronger for long-horizon tasks. How should I infer this from the mathematical result within the proposition?”**
>
> Proposition 3.2 states that the guidance gap scales at least proportionally to the square root of the dimensionality $d$ of the trajectory representation:
>
> $$\Delta_{\mathrm{guidance}}\bigl(\boldsymbol{\tau}^i\bigr) \ge \frac{\,c\,}{\sqrt{1-\alpha_i}}\sqrt{d}$$
>
> We clarify that the dimensionality $d$ here directly relates to the length of the planning horizon. Specifically, in diffusion-based planning, trajectories are represented by concatenating states and actions across the entire planning horizon. Therefore, the dimension $d$ grows linearly with the length of the planning horizon.
> Consequently, the lower bound provided by Proposition 3.2 indicates that as the planning horizon increases, the guidance gap inevitably grows, which implies a higher likelihood and severity of manifold deviation in longer-horizon tasks.
>
> - **“My main concern regarding the paper regards the risk of ending up obtaining a sampler excessively regularized from offline data as explained within my point (2) of Claims And Evidence section.”**
>
> Thank you for raising this important point. We acknowledge that our approach inherently regularizes trajectories towards the offline data manifold. However, our primary objective in this work is to ensure the generation of feasible, high-return trajectories for offline RL tasks particularly in settings where safety and reliability are crucial. This highlights a fundamental trade-off between trajectory feasibility and exploration freedom.
>
> To clarify this explicitly, we will revise the manuscript to discuss this limitation in detail, clearly identifying scenarios where LoMAP provides the most substantial benefits (e.g., safety-critical domains or tasks prioritizing feasibility over aggressive exploration).
>
> Furthermore, we believe LoMAP can be effectively combined with trajectory stitching and data augmentation techniques [2, 3], which enrich offline datasets with diverse, synthetic trajectories, which we leave for future work.
>
> References:
>
> [1] Um et al., Don't Play Favorites: Minority Guidance for Diffusion Models. In ICLR, 2024.
>
> [2] Li et al., DiffStitch: Boosting Offline Reinforcement Learning with Diffusion-based Trajectory Stitching. In ICML, 2024.
>
> [3] Chen et al., Extendable Long-Horizon Planning via Hierarchical Multiscale Diffusion. arXiv, 2025.

---

### Official Review · Reviewer_wazP · 2025-03-14

**Overall Recommendation:** 4

**Summary:**

The authors tackle the problem of approximate energy guidance with diffusion policies in the context of offline RL. Partial energies naively trained through MSE loss such as in Diffuser are only lower bounds to the true energy, and so using their gradients for conditional sampling can push trajectories off the manifold supported by the offline dataset. The authors propose a simple to implement technique to project the latent trajectories back into a low rank approximation of the K nearest neighbors to the dataset, noised to the current diffusion timestep. The K nearest neighbors are obtained with the expected denoised trajectory from the current diffusion step using Tweedie's formula. The authors demonstrate that using this correction improved performance across tasks in the D4RL suite and specifically highlight generation of valid trajectories in Maze2D and AntMaze tasks, where Diffuser often generates infeasible paths that pass through walls.

**Claims And Evidence:**

The claims in the paper are clear, and evidence is convincing

**Essential References Not Discussed:**

I think it would improve the paper to include references (and perhaps even a comparison) to asymptotically unbiased guidance strategies such as Twisted Diffusion Sampler (TDS) [1] and some other SMC based methods listed in the above survey paper, and broadly discuss this line of work (perhaps in the appendix).

[1] Practical and Asymptotically Exact Conditional Sampling in Diffusion Models, https://arxiv.org/abs/2306.17775

**Experimental Designs Or Analyses:**

The experiments seem to be sound, I did not notice any specific issue.

**Methods And Evaluation Criteria:**

D4RL, specifically the maze tasks make sense for evaluation of this approach. However, I will note that D4RL is a very saturated benchmark, and going forward I hope OGBench is used instead. Since this work must have been done concurrently with the introduction of OGBench, I do not consider this a weakness.

**Other Comments Or Suggestions:**

Could you include QGPO (from the contrastive energy prediction paper) to the results tables? It is a technique which trains a time dependent energy model for unbiased guidance, so perhaps can serve as a sort of upper bound in performance.

**Other Strengths And Weaknesses:**

### Strengths
1. The paper is clearly written, with useful illustrations.
2. The proposed method is novel to my understanding.

### Weaknesses
1. K-nearest neighbors computation with the entire dataset could be expensive, especially with growth in dimensionality of states and trajectory length. This is especially problematic for any sort of real world control tasks.
2. The approach used to compute K-nearest neighbors from the dataset, with Tweedie's denoised estimate as the anchor and cosine distance the metric function may not be good heuristics for other problems (especially with high dimensional state spaces). Perhaps some kind of abstract trajectory latent representations could be used instead?

**Questions For Authors:**

1. With the IVF method for faster nearest neighbors as described in Appendix D, what was the wall clock time to generate a single trajectory for a task like AntMaze? How expensive is the procedure with naive nearest neighbors?

**Relation To Broader Scientific Literature:**

The paper is relevant to the application of diffusion models for offline RL, which is an active area of research. Contrastive Energy Prediction [1] and Relative Trajectory Balance are quite relevant to this work, as hey similarly focus on the problem of inaccurate energy guidance in the context of offline RL. However in that paper, the authors introduce a contrastive training strategy to learn an unbiased guidance function, whereas this paper instead introduces a simpler to implement (but perhaps less general) training-free guidance method. The problem of training-free approximate/asymptotically unbiased guidance for diffusion models is heavily explored outside the context of offline RL, as explored in this survey paper [3].

[1] Contrastive Energy Prediction for Exact Energy-Guided Diffusion Sampling in Offline Reinforcement Learning, https://arxiv.org/abs/2304.12824

[2] Amortizing intractable inference in diffusion models for vision, language, and control, https://arxiv.org/abs/2405.20971

[3] Inference-Time Alignment in Diffusion Models with Reward-Guided Generation: Tutorial and Review, https://arxiv.org/abs/2501.09685

**Theoretical Claims:**

The theoretical claims are valid, and I checked the only major proof in Appendix A.

---

> ### Author Rebuttal · Authors · 2025-03-30
>
> Thank you for the positive review and constructive feedback. Please find our detailed response below.
>
> - **“With the IVF method for faster nearest neighbors as described in Appendix D, what was the wall clock time to generate a single trajectory for a task like AntMaze? How expensive is the procedure with naive nearest neighbors?”**
>
> Thank you for highlighting this important practical aspect. We measured the wall-clock time required by Hierarchical Diffuser (HD) with LoMAP in the AntMaze task, averaging the generation time per trajectory using a single NVIDIA A10G GPU, with the number of neighbors set to $K=10$.
>
> | **Method** | **Total wall-clock time per trajectory (sec)** | **KNN-search time (sec)** |
> | --- | --- | --- |
> | Naive KNN | 21.62 | 21.02 |
> | IVF-based approximate KNN (**ours**) | 0.53 | 0.01 |
>
> As shown above, the naive KNN search is prohibitively expensive, requiring over 21 seconds to generate a single trajectory, with the vast majority of time dedicated to the KNN search itself. In contrast, our IVF-based approximate KNN significantly reduces runtime, with only a negligible fraction of the total time spent on the nearest neighbor search.
>
> - **“Could you include QGPO to the results tables? It is a technique which trains a time dependent energy model for unbiased guidance, so perhaps can serve as a sort of upper bound in performance.”**
>
> Thank you for suggesting QGPO as an additional baseline. We agree that including QGPO can indeed serve as a valuable upper-bound comparison for performance. We appreciate this insightful recommendation and will incorporate the QGPO results into the final version of our paper.
>
> - “**Cosine distance metric function may not be good heuristics for other problems (especially with high dimensional state spaces).**”
>
> As the reviewer correctly points out, cosine distance may not be suitable for certain high-dimensional state spaces, especially pixel-based environments. Exploring effective manifold approximation methods for more complex, high-dimensional state spaces remains an important direction for future work. We believe one promising approach would be to combine LoMAP with latent trajectory representations [1, 2]. We will clarify this limitation explicitly in the final version.
>
> - **"An additional references to asymptotically unbiased guidance strategies (perhaps in the appendix)"**
>
> Thank you for this valuable suggestion. We will discuss asymptotically unbiased guidance strategies further in the related work section of the final version.
>
> References
>
> [1] Co-Reyes et al., Self-consistent trajectory autoencoder:
> Hierarchical reinforcement learning with trajectory embeddings. In ICML, 2018.
>
> [2] Jiang et al., Efficient Planning in a Compact Latent Action Space. In ICLR, 2023.

---

> > ### Comment · Reviewer_wazP · 2025-04-04
> >
> > Thank you for the response, I will maintain my score 4 (accept).

---

> > > ### Author Response · Authors · 2025-04-05
> > >
> > > We sincerely appreciate your time and dedication to providing us with your valuable feedback.

---

### Official Review · Reviewer_AXnQ · 2025-03-14

**Overall Recommendation:** 4

**Summary:**

Classifier guidance can introduce distribution shift during diffusion sampling. This paper proposes a training-free method to constrain guided diffusion within a learned manifold by projecting noisy samples onto a local low-dimensional manifold, approximated using nearest neighbors from the training set at each diffusion step.

**Claims And Evidence:**

The claim that guided diffusion can lead to manifold deviation is supported by theoretical analysis. However, the assertion that LoMAP, the proposed projection method, effectively bridges the guidance gap is less convincing, as evidenced by its lack of significant improvement over baselines like DD and TAT in Table 2.

**Essential References Not Discussed:**

Many recent works have explored diffusion sampling that satisfies constraints [1-3]. In particular, [1] introduced a training-free, plug-and-play four-line modification to guided diffusion that directly addresses the issue shown in your Figure 3. It also enables minority guidance without introducing constraint violations, as demonstrated in your Figure 5. Comparing your proposed method against this simpler alternative would help verify whether it offers a meaningful improvement over existing approaches in addressing the same challenge.

[1] Inference-Time Policy Steering through Human Interactions
[2] Learning Diverse Robot Striking Motions with Diffusion Models and Kinematically Constrained Gradient Guidance
[3] DISCO: Language-Guided Manipulation with Diffusion Policies and Constrained Inpainting

**Experimental Designs Or Analyses:**

Yes.

**Methods And Evaluation Criteria:**

Yes.

**Other Comments Or Suggestions:**

Figures 1(a) and 1(b) do not appear significantly different, making it difficult to highlight the differences. Additionally, Figure 1(c) does not clearly convey the idea of your method. It may be helpful to refine the figure to better illustrate the key improvements and make the distinctions more visually compelling.

**Other Strengths And Weaknesses:**

Strength – The paper provides a theoretical demonstration of manifold deviation.

Weakness – The results do not show a strong improvement over chosen baselines such as DD or TAT. Additionally, the study lacks comparisons with directly relevant baselines, as mentioned above.

**Questions For Authors:**

1. What is the intuition behind the idea that projecting onto the subspace approximated by PCA of nearest neighbors will reduce manifold deviation during diffusion? Why does PCA, in particular, help? Does the number of PCA components impact performance?

2. Could this projection (an approximation) at later diffusion steps introduce constraint violations? Have you experimented with applying LoMAP only during the early diffusion steps or, conversely, only during the later diffusion steps, rather than applying it across all diffusion steps?

3. What is the computational overhead of finding nearest neighbors from a large offline trajectory dataset at each diffusion step? Does your algorithm require access not only to the diffusion model but also to the dataset the model was trained on?

**Relation To Broader Scientific Literature:**

See below the essential reference section.

**Theoretical Claims:**

Yes.

---

> ### Author Rebuttal · Authors · 2025-03-31
>
> Thank you for the detailed review and constructive feedback on our work. Please find our detailed responses below.
>
> - **Not a strong improvement over chosen baselines**
>
> We appreciate the valuable feedback provided by the reviewer. However, we respectfully disagree with the subjective judgment that there was “not strong” improvement. We believe the performance improvements demonstrated by our method are meaningful. Specifically, our approach consistently outperforms all baselines in terms of average performance across three established benchmarks, despite the simplicity of our method.
>
> - **Missing baselines**
>
> We sincerely thank the reviewer for pointing out several interesting works. However, we would like to clarify why the suggested baselines are not directly suitable for comparison in our setting:
>
> - **[1]** assumes human-provided guidance in the form of sketches or demonstrations, which fundamentally differs from our scenario where no such human intervention is assumed. Additionally, [1] relies on a diffusion-based policy rather than a diffusion-based planner, making direct comparisons inappropriate. The reviewer's comment, "It also enables minority guidance without introducing constraint violations, as demonstrated in your Figure 5," is also not entirely accurate. The trajectories in [1] were generated using a diffusion policy trained with random walks rather than being goal-conditioned as in our experiments.
> - **[2]** assumes the availability of differentiable constraints. For instance, applying [2] directly to the maze environment would require prior knowledge about the exact location of walls, making the comparison unfair.
> - **[3]** focuses specifically on manipulation scenarios guided by language instructions, which significantly deviates from the tasks and settings considered in our work, making it an unsuitable baseline.
>
> ---
> - **Figure suggestion**
>
> Thank you for the suggestions! We will incorporate them into the final version of our paper.
>
> - **What is the intuition behind the LoMAP**
>
> Please see response for reviewer XEyQ for the details.
>
> - **Impact of the number of PCA components**
>
> We have conducted additional experiments by varying the number of PCA components ($K$). As shown in the results below, LoMAP demonstrates robustness provided that K is above a certain threshold. We select $K=10$ as it offers computational efficiency comparable to $K=20$, yet achieves similar performance.
>
> | Environment | **Diffuser** | **K=5** | **K=10** | **K=20** |
> | --- | --- | --- | --- | --- |
> | halfcheetah-med-expert | 88.9 ± 0.3 | 91.2 ± 0.3 | 91.1 ± 0.2 | 90.9 ± 0.2 |
> | hopper-med-expert | 103.3 ± 1.3 | 108.9 ± 2.9 | 110.6 ± 0.3 | 110.9 ± 0.2 |
> | walker2d-med-expert | 106.9 ± 0.2 | 107.8 ± 0.2 | 109.2 ± 0.1 | 108.9 ± 0.1 |
> | halfcheetah-med | 42.8 ± 0.3 | 44.5 ± 0.1 | 45.4 ± 0.1 | 44.9 ± 0.1 |
> | hopper-med | 74.3 ± 1.4 | 90.16 ± 2.0 | 93.7 ± 1.5 | 94.1 ± 1.7 |
> | walker2d-med | 79.6 ± 0.6 | 79.17 ± 1.8 | 79.9 ± 1.2 | 81.4 ± 1.1 |
> | halfcheetah-med-replay | 37.7 ± 0.5 | 38.2 ± 1.2 | 39.1 ± 1.0 | 38.9 ± 0.7 |
> | hopper-med-replay | 93.6 ± 0.4 | 96.4 ± 0.2 | 97.6 ± 0.6 | 96.3 ± 0.1 |
> | walker2d-med-replay | 70.6 ± 1.6 | 74.7 ± 3.1 | 78.7 ± 2.2 | 78.9 ± 1.8 |
> - **“Have you tried applying LoMAP only in early steps or only in later steps?”**
>
> Yes, we explored applying LoMAP exclusively in either early or later stages of the diffusion sampling process. We observed that applying LoMAP only in early steps was not particularly effective. In contrast, applying LoMAP during intermediate to later diffusion steps yielded the most significant improvements. This indicates that the mismatch between the model’s reverse transition and the true data distribution becomes most pronounced during these intermediate steps, aligning with similar observations reported in prior study [4].
>
> - **“Computational overhead of finding nearest neighbors”**
>
> Please see response for reviewer wazP for the details.
>
> - **“Does your algorithm require access not only to the diffusion model but also to the dataset the model was trained on?”**
>
> LoMAP, as currently presented, assumes the existence of an offline dataset, consistent with other offline RL baselines. For instance, Decision Diffuser (DD) retrains a conditional diffusion planner from scratch using offline datasets, and RGG trains an out-of-distribution (OOD) score predictor utilizing offline datasets. Unlike these methods, our approach leverages the dataset directly without additional training, making it simple and straightforward to apply.
>
> References:
>
> [1] Shi et al., Inference-Time Policy Steering through Human Interactions. In ICRA, 2025.
>
> [2] Lee et al., Learning Diverse Robot Striking Motions with Diffusion Models and Kinematically Constrained Gradient Guidance. arXiv, 2024.
>
> [3] Hao et al., DISCO: Language-Guided Manipulation with Diffusion Policies and Constrained Inpainting. arXiv, 2024.
>
> [4] Na et al., Diffusion Rejection Sampling. ICML, 2024.

---

> > ### Comment · Reviewer_AXnQ · 2025-04-07
> >
> > I appreciate the authors' rebuttal effort.
> >
> > However, I respectfully disagree with the authors' claim that their approach "consistently outperforms all baselines." If you look at individual entries in Table 2, baselines DD, TAT, and RGG either outperform LoMAP or are within the error bars of LoMAP. Admittedly, LoMAP has the best overall average performance, but there are no error bars, and DD, TAT, and RGG's average performances come very close. Therefore, objectively, I cannot agree that the proposed LoMAP provides a strong improvement over the baselines.
> >
> > Furthermore, the majority of baselines (BC, CQL, IQL, DT, TT, MOPO, MOReL, DD) do not natively address the issue of inadvertently sampling infeasible plans during conditional generation. Therefore, surpassing their performance by a large margin cannot be used as evidence that LoMAP improves upon the SoTA methods for mitigating distribution shift (or for sampling feasible plans that satisfy constraints) during conditional generation. Rather, the authors should focus their efforts on meaningfully beating TAT, RGG, and the three works [1–3] I mentioned in this spirit.
> >
> > The authors dismissed [1] by saying it assumes human guidance, which is not true. [1] only requires human guidance when performing conditional generation—just as LoMAP uses goal conditions to do conditional plan generation. In fact, one could replace human guidance with goal conditioning of your tasks, and the same algorithm would apply. Given that [1] is just a four-line algorithmic change to Diffuser (which the authors used) that reduces infeasible plans during conditional generation, it is worth comparing to LoMAP (which appears to be a much more complicated algorithm).
> >
> > The authors dismissed [2] by saying applying [2] directly to the maze environment would require prior knowledge about the exact location of walls. This does not seem overly restrictive, given that LoMAP requires the entire offline dataset to be available in order to find nearest neighbors during deployment-time planning. In fact, if one already has access to the entire offline dataset, one can simply recover the maze walls from the offline trajectory dataset and plug in [2] to see whether LoMAP truly performs better at sampling plans that satisfy constraints.
> >
> > The authors dismissed [3] by saying [3] focuses specifically on manipulation scenarios guided by language instructions, which is not relevant. Admittedly, [3] includes components that are irrelevant to LoMAP. However, one core innovation of [3] is to perform simple gradient descent during diffusion so that sampling falls back onto the data manifold, thereby reducing the sampling of infeasible plans ([3], Fig. 3(b)). This practically works well and appears much simpler than the proposed LoMAP. Therefore, it is worth comparing LoMAP to [3] to evaluate whether the added complexity of LoMAP's algorithmic design truly brings additional benefits.
> >
> > Lastly, I am also concerned that many of the results rely on normalized average returns (in line with offline RL works) as a performance measure, which might not be the most appropriate metric for evaluating whether a sampled plan is feasible. The authors may consider using binary counts, such as their proposed artifact ratios, for all experiments to better demonstrate LoMAP's improvement in sampling feasible plans.
> >
> > While I really appreciate the cleverness of the algorithmic design of LoMAP and the thorough theoretical analysis, I am concerned about the practical value of the algorithm. Given that the authors' rebuttal glosses over my concern about whether LoMAP truly improves upon the latest methods that reduce infeasible plans during conditional diffusion sampling, I will lower my score to reject.
> >
> > [1] Inference-Time Policy Steering through Human Interactions [2] Learning Diverse Robot Striking Motions with Diffusion Models and Kinematically Constrained Gradient Guidance [3] DISCO: Language-Guided Manipulation with Diffusion Policies and Constrained Inpainting

---

> > > ### Author Response · Authors · 2025-04-08
> > >
> > > We sincerely appreciate your additional feedback. Below, we present additional comparisons and experiments designed to address the concerns, specifically regarding (1) direct comparisons with [1--3], and (2) metrics for feasibility.
> > >
> > > * **Additional Comparisons with [1], [2], and [3]**
> > >
> > > Following the reviewer's feedback, we conducted additional experiments in the Maze2D-Large environment to evaluate the effectiveness of our LoMAP method compared to [1], [2], and [3]. For a fair comparison, we used the same diffusion model as described in our paper (e.g., using a planning horizon four times longer than in [1], including velocity states).
> > >
> > > **Comparison with [1]:** As suggested, we adapted the method from [1] by goal-conditioning via stochastic sampling, tuning the number of MCMC sampling steps $(2, 4, 6, 8)$.
> > >
> > > **Comparison with [2]:** We approximated walls as multiple spheres in Maze2D and defined a sphere-based constraint cost following [4]. Specifically:
> > >
> > > $$
> > > J_{c}(\tau)
> > > = \sum_{m=1}^{M}
> > >   \sum_{t=1}^{H}
> > >     \max \Bigl(
> > >         r
> > >         -
> > >         \mathrm{dist}\bigl(\tau_{t}, p_{m}\bigr),
> > >      0 \Bigr),
> > > $$
> > >
> > > where $H$ is the planning horizon, $p_m$ is the center of the sphere, and $r$ is its radius. We tuned the guidance scale within the range (0.001, 0.01, 0.05, 0.1).
> > >
> > > **Comparison with [3]:** To emulate VLM-based keyframe generation akin to [3], we trained a high-level policy that learns the optimal $k$-steps jump using Hierarchical Implicit Q-learning (HIQL) [5] to generate subgoals. These subgoals served as keyframes for the inpainting optimization technique from [3], with $k=25$ following the official implementation in ogbench.
> > >
> > > We first extended the experiments from Section 4.1 to compare artifact ratios:
> > >
> > > | # of plans | **LoMAP (ours)** | **Diffuser** | **[1]** | **[2]** | **[3]** |
> > > | --- | --- | --- | --- | --- | --- |
> > > | 10 | **0.35** | 0.50 | 0.42 | 0.49 | 0.43 |
> > > | 20 | **0.35** | 0.62 | 0.44 | 0.54 | 0.46 |
> > > | 30 | **0.38** | 0.66 | 0.47 | 0.61 | 0.49 |
> > >
> > > LoMAP consistently outperformed all compared methods, significantly reducing the artifact ratio. Notably, even [2], despite leveraging exact wall constraints, failed to match LoMAP's effectiveness. We speculate that the nonconvex nature of the constraints makes gradient updates inadequate for reliably projecting trajectories into collision-free regions. Additionally, although stochastic sampling [1] and inpainting optimization [3] improved over the standard Diffuser, they still exhibited higher artifact ratios compared to LoMAP.
> > >
> > > Below, we report the normalized return for Maze2D-Large and Multi2D-Large. LoMAP significantly outperforms [1], [2], and [3] in terms of normalized average returns, further validating the effectiveness of our approach.
> > >
> > > |  | **LoMAP (ours)** | **Diffuser** | **[1]** | **[2]** | **[3]** |
> > > | --- | --- | --- | --- | --- | --- |
> > > | Maze2D Large | 151.9 ± 2.7 | 123.0 ± 6.4 | 135.1 ± 4.0 | 129.0 ± 5.3 | 137.9 ± 2.4 |
> > > | Multi2D Large | 154.7 ± 0.3 | 132.1 ± 5.8 | 143.8 ± 4.7 | 141.3 ± 4.3 | 145.6 ± 3.1 |
> > >
> > >
> > > * **Appropriate metric for feasibility**
> > >
> > > We acknowledge the limitation that artifact ratios can be directly measured only in environments like maze domain, where explicit constraints are easily defined. Therefore, we further evaluated feasibility in locomotion tasks using the dynamic mean squared error (dynamic MSE), measuring how closely generated trajectories adhere to true environment dynamics:
> > >
> > > $$\text{dynamic MSE}=||f^*(s,a)-s'||^2,$$
> > >
> > > where $f^*$ denotes the true dynamics model. We generated 100,000 trajectory samples for each environment and reported the average dynamic MSE.
> > >
> > > | Environment | **w/o LoMAP** | **w/ LoMAP** |
> > > | --- | --- | --- |
> > > | halfcheetah-med-expert | 0.363 | **0.295** |
> > > | hopper-med-expert | 0.027 | **0.020** |
> > > | walker2d-med-expert | 0.391 | **0.293** |
> > > | halfcheetah-med | 0.352 | **0.285** |
> > > | hopper-med | 0.024 | **0.021** |
> > > | walker2d-med | 0.395 | **0.293** |
> > > | halfcheetah-med-replay | 0.710 | **0.555** |
> > > | hopper-med-replay | 0.049 | **0.045** |
> > > | walker2d-med-replay | 0.829 | **0.506** |
> > >
> > > Consistently, LoMAP significantly reduced dynamic MSE, clearly demonstrating its effectiveness in generating trajectories that better adhere to environmental dynamics.
> > >
> > > We again appreciate your time and dedication to providing us with your valuable feedback.
> > >
> > > References:
> > >
> > > [1] Shi et al., Inference-Time Policy Steering through Human Interactions. In ICRA, 2025.
> > >
> > > [2] Lee et al., Learning Diverse Robot Striking Motions with Diffusion Models and Kinematically Constrained Gradient Guidance. arXiv, 2024.
> > >
> > > [3] Hao et al., DISCO: Language-Guided Manipulation with Diffusion Policies and Constrained Inpainting. arXiv, 2024.
> > >
> > > [4] Shaoul et al., Multi-Robot Motion Planning with Diffusion Models. ICLR, 2025.
> > >
> > > [5] Park et al., HIQL: Offline Goal-Conditioned RL with Latent States as Actions. ICML, 2023.

---

### Official Review · Reviewer_XEyQ · 2025-03-17

**Overall Recommendation:** 4

**Summary:**

This paper addresses the limitation in diffusion-based trajectory planning for RL tasks. Previous works in diffusion models often produce infeasible trajectories due to "manifold deviation" during the sampling process, so the authors proposed a novel method LoMAP, which is a training-free framework. It projects diffusion model samples onto locally approximated low-rank manifolds derived from offline datasets to prevent infeasible trajectory generation caused by guidance errors during the sampling process. The experiments on Gym achieved good results compared to the baseline.

**Claims And Evidence:**

The authors make a clear point on the problem of the guidance gap in diffusion models and provide theoretical proof to understand the issue. However, I'm a bit lost on why KNN and PCA can solve this problem, and I feel section 3.2 is a bit isolated from the discussion of manifold deviation. I hope the authors can explain more about this part and expand this section since it is the key part of this paper.

**Essential References Not Discussed:**

Although this paper focused on diffusion models, I suggest the author include more related works based on Decision Transformers (DT) as baseline or related works, especially those with classifier or classifier-free guidance in planning. Here are some papers:
[1]: Latent Plan Transformer for Trajectory Abstraction: Planning as Latent Space Inference
[2]: Q-learning Decision Transformer: Leveraging Dynamic Programming for Conditional Sequence Modelling in Offline RL

**Experimental Designs Or Analyses:**

The experimental designs follow the most commonly used environment and baselines.

This whole paper talks about the manifold projection, and I'm wondering if the author can provide some evidence this method actually helps with more accurate manifold approximation.

**Methods And Evaluation Criteria:**

The method and analysis are clear and simple, which is a plus. The whole framework is lightweight, but I do not fully understand why LoMAP can help with the issue of diffusion sampling.

The experiment part seems solid and the performance is good. I suggest adding more DT-related baseline methods. See the below sections for references.

**Other Comments Or Suggestions:**

N/A

**Other Strengths And Weaknesses:**

I like to narrative of the paper. The weaknesses are discussed above.

**Questions For Authors:**

See the above sections.

**Relation To Broader Scientific Literature:**

N/A

**Theoretical Claims:**

The theoretical analysis in section 3.1 looks good to me. And the introduction of Diffuser and related literature is clear and concise.

---

> ### Author Rebuttal · Authors · 2025-03-30
>
> Thank you for the detailed review and constructive feedback on this work. We especially appreciate the insightful questions regarding the accurate manifold approximation. Please find our detailed answers below.
>
> - **"I do not fully understand why LoMAP can help with the issue of diffusion sampling."**
>
> We thank the reviewer for this important question. To clarify, the core intuition behind LoMAP is grounded in the observation that feasible trajectories from offline datasets lie on an intrinsically low-dimensional manifold embedded within a high-dimensional trajectory space. Diffusion-based sampling methods can deviate from this manifold due to inaccuracies in guidance (as analyzed in Section 3.1), leading to infeasible trajectories.
>
> LoMAP addresses this problem by iteratively projecting guided diffusion samples back onto an approximated local manifold derived directly from offline trajectories. Specifically, at each diffusion step, we first compute a denoised estimate of the current trajectory sample using Tweedie's formula. We then retrieve the $k$-nearest neighbors from the offline dataset, which naturally represent trajectories that closely adhere to the true data manifold. By forward-diffusing these neighbor trajectories to the current diffusion timestep, we approximate the local structure of the intermediate manifold ($\mathcal{M}_{i-1}$).
>
> PCA then provides a convenient and effective way to approximate the local manifold around this neighborhood. Because PCA identifies principal directions of variance, it naturally captures the major local geometric structures of the feasible manifold represented by nearby trajectories. By projecting the diffusion sample onto this PCA-derived subspace, we remove the off-manifold components, effectively correcting the artifact trajectories.
>
> - **"I'm wondering if the author can provide some evidence this method actually helps with more accurate manifold approximation."**
>
> Thanks for the important question. To provide further evidence, we computed the Realism Score [1], which quantifies how closely generated trajectories lie to the true manifold defined by the offline dataset. Specifically, we approximated the true manifold using k-NN hyperspheres constructed from 20,000 offline trajectories and generated 100,000 trajectory samples to measure the average realism score. As shown below, applying LoMAP consistently yields higher realism scores compared to diffusion sampling without LoMAP, clearly demonstrating that our method effectively produce samples closer to the true data manifold.
>
> | Environment | **w/o LoMAP** | **w/ LoMAP** |
> | --- | --- | --- |
> | Maze2D U-Maze | 1.23 | **1.30** |
> | Maze2D Medium | 1.40 | **1.56** |
> | Maze2D Large | 1.36 | **1.47** |
>
> - **“I suggest adding more DT-related baseline methods.”**
>
> Thank you for suggesting additional DT-related baselines. Following your advice, we now include comparisons with QDT [2], LPT [3] (guidance-based methods), and WT [4] (waypoint-based method). For MuJoCo locomotion tasks, we used the QDT implementation from the d3rlpy library and the official implementation provided by the authors for LPT, while WT results are taken directly from the original paper [4]. As shown in the table below, despite its simplicity, LoMAP consistently achieves the best average performance across these benchmarks. We will incorporate these additional comparisons into the final version of the paper.
>
> | Environment | **QDT** | **LPT** | **WT** | **ours** |
> | --- | --- | --- | --- | --- |
> | halfcheetah-med-expert | 89.8 ± 0.7 | 90.8 ± 0.19 | **93.2 ± 0.5** | 91.1 ± 0.23 |
> | hopper-med-expert | 109.4 ± 2.3 | **111.4 ± 0.31** | 110.9 ± 0.6 | 110.6 ± 0.29 |
> | walker2d-med-expert | 108.8 ± 0.7 | 109.1 ± 0.04 | **109.6 ± 1.0** | 109.2 ± 0.05 |
> | halfcheetah-med | 42.3 ± 0.4 | 43.5 ± 0.08 | 43.0 ± 0.2 | **45.4 ± 0.13** |
> | hopper-med | 66.5 ± 6.3 | 63.8 ± 1.47 | 61.1 ± 1.4 | **93.7 ± 1.54** |
> | walker2d-med | 67.1 ± 3.2 | **81.1 ± 0.33** | 74.8 ± 1.0 | 79.9 ± 1.21 |
> | halfcheetah-med-replay | 35.6 ± 0.5 | **40.7 ± 0.12** | 39.7 ± 0.3 | 39.1 ± 0.99 |
> | hopper-med-replay | 52.1 ± 20.3 | 89.9 ± 0.61 | 88.9 ± 2.4 | **97.6 ± 0.58** |
> | walker2d-med-replay | 58.2 ± 5.1  | 75.7 ± 0.34 | 67.9 ± 3.4 | **78.7 ± 2.2** |
> | **average** | 69.9 | 78.4 | 76.8 | **82.8** |
>
> References
>
> [1] Kynkäänniemi et al., Improved precision and recall metric for assessing generative models. In NeurIPS, 2019.
>
> [2] Yamagata et al., Q-learning Decision Transformer: Leveraging Dynamic Programming for Conditional Sequence Modelling in Offline RL. In ICML, 2023.
>
> [3] Kong et al., Latent Plan Transformer for Trajectory Abstraction: Planning as Latent Space Inference. In NeurIPS, 2024.
>
> [4] Badrinath et al., Waypoint Transformer: Reinforcement Learning via
> Supervised Learning with Intermediate Targets. In NeurIPS, 2023.

---

### Decision · Program_Chairs · 2025-05-01

**Decision:**

Accept (poster)

**Comment:**

Diffusion-based generative models for RL have inconsistent reliability due to stochastic risk of infeasible trajectories. Via identifying inaccurate sampling guidance as the cause and showing manifold deviation, the paper proposes LoMAP, a training-free method projecting guided samples onto a low-rank subspace from offline data to prevent such infeasible trajectories. I think the paper addresses a key problem in diffusion-based RL methods and will be quite helpful for further research in this direction. Thus I recommend an acceptance for this paper.